# Rapid evolution in necromass use under resource limitation reduces persistence in producer-decomposer microbial biospheres
Shaoyi Jiang(姜少毅) [1,2], Alexandra Halbleib[1], Janis Antonovics [3,4], Mitja Remus-Emsermann [1], Matthias C. Rillig [1,2] ✉ & India Mansour [1,2] ✉

Decomposer bacteria recycle necromass, the organic remnants of dead cells. This process underlies ecological system persistence via provisioning matter for future generations; however, it is unknown if more efficient necromass recycling also improves the persistence of the microbial ecosystem reliant upon it. Here, we utilize severely resource-limited conditions to force a decomposer bacterium, *Escherichia coli,* to grow using only its own necromass and subsequently use these adapted populations to test if improved necromass recycling ability contributes to longer persistence times of closed microbial biospheres. *E. coli* strain MC4100 was incubated in matter-closed systems in an isotonic saline solution without added nutrients for 60 days in spatially homogeneous or heterogeneous conditions. We observed phenotypic adaptive changes in *E. coli* populations, including improved growth on various carbon sources, loss of motility, and enhanced biofilm formation, which were specific to the spatial structure conditions. While these changes were beneficial for *E. coli* in monoculture, during co-culture with the alga *Chlamydomonas reinhardtii*, biospheres containing the adapted populations exhibited a 25% reduction in persistence under spatially heterogeneous conditions compared to those containing non-adapted *E. coli* populations. While nutrient cycling must occur for life in biospheres to self-sustain, enhanced recycling of necromass may actually undermine biosphere persistence.

Microorganisms frequently encounter nutrient scarcity across diverse environments[1–5]. To survive, they employ strategies such as altering morphology[6,7], modulating motility[8], enhancing adhesion[9], simplifying metabolic pathways[10], or entering dormancy[11] and growth arrest[12]. Notably, many bacteria persist for decades in energy-limited environments by recycling necromass, the organic remnants of dead cells[13]. In *Escherichia coli*, long-term stationary phase (LTSP) populations endure extreme nutrient depletion through density-dependent death and cellular debris recycling[14]. This process involves continuous adaptation characterized by genetic parallelism[15], mutations in global regulators (e.g., *rpoS*)[16], and a shift toward slow growth and stress resistance[17], often resulting in a growth advantage in stationary phase (GASP) phenotype[18]. Despite these known survival mechanisms, the ecological consequences of starvation-induced adaptation, particularly its impact on interspecies interactions and ecosystem stability, are underexplored and unclear. Furthermore, the interplay between spatial structure (critical for promoting genetic diversity[19], species coexistence[20], and ecosystem sustainability[21]) and microbial adaptation under prolonged resource scarcity requires further exploration.

To investigate microbial ecosystem dynamics under controlled, resource-limited conditions, we employed microbial biospheres[21,22]. These experimental systems allow laboratory-scale observation of system failure, which we defined as the collapse of primary production and therefore ongoing carbon cycling, reflected by the loss of photosynthetic activity[22]. Specifically, these biospheres are matter-closed but energy-open (light), necessitating internal carbon and nutrient cycling. This matter-closure minimizes external interference, such as species immigration, and requires

[1]Institut für Biologie, Freie Universität Berlin, Berlin, Germany. [2]Berlin-Brandenburg Institute of Advanced Biodiversity Research, Berlin, Germany. [3]Department of Biology, University of Virginia, Charlottesville, VA, USA. [4]Royal Botanical Gardens Edinburgh, Edinburgh, UK. ✉e-mail: rillig@zedat.fu-berlin.de; india.mansour@gmail.com; india.mansour@fu-berlin.de

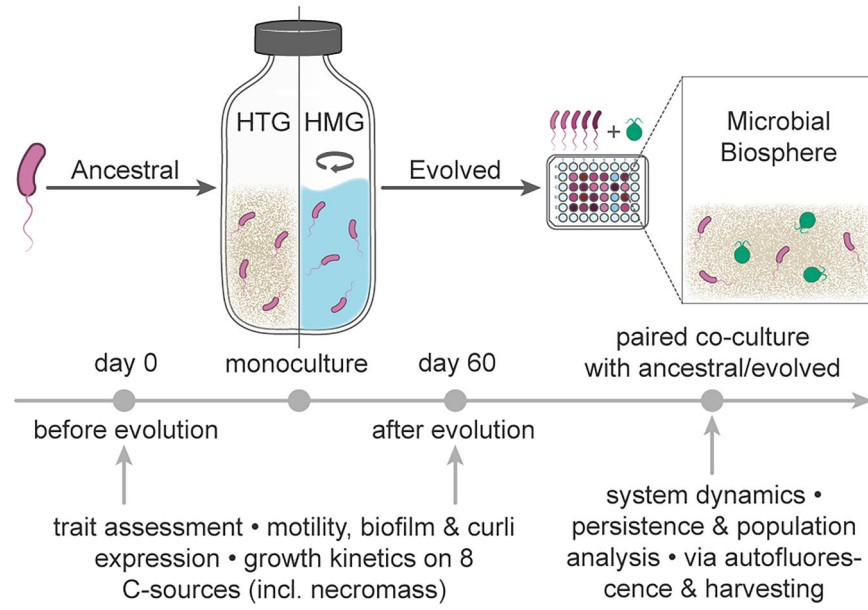

**Fig. 1 | Schematic of the two-step experimental design and workflow.** Wild-type *Escherichia coli* (WT *E. coli*) was first subjected to 60-day mono-culture in two enclosed environments without added nutrients (HTG: spatially heterogeneous; HMG: spatially homogeneous). The evolved populations, along with the WT progenitor, were then assayed for motility, biofilm formation, pellicle formation, curli fimbriae expression, and growth curves on eight distinct carbon sources. Subsequently, each evolved population and the WT progenitor were individually paired with *Chlamydomonas reinhardtii* for co-culture in a spatially structured microbial biosphere established in 48-well plates. System persistence was tracked by continuous measurement of chlorophyll auto-fluorescence, and population dynamics were investigated through four destructive samplings.

microbial regulation of the atmosphere for persistence. Unlike open systems like chemostats[23], microbial biospheres lack nutrient inputs, requiring the evolution of self-contained nutrient cycles. Standardized replication of the biospheres allows for precise monitoring to distinguish deterministic trends from stochastic fluctuations[23]. To mimic the core carbon cycle, we co-cultured the autotroph *Chlamydomonas reinhardtii*[24] and decomposer *E. coli* MC4100. *E. coli* metabolizes necromass, releasing $CO_2$ for photosynthetic assimilation by *C. reinhardtii*, which in turn maintains aerobic conditions via $O_2$ release. Interactions may also include mixotrophy, as *C. reinhardtii* may utilize *E. coli* metabolites such as acetate[25]. System persistence is monitored at the ecosystem level via photosynthetic activity. Accordingly, we consider the system to have failed once this activity ceases, regardless of temporary species persistence on alternative substrates.

Although nutrient recycling is essential for self-sustaining biospheres[22], whether increased recycling efficiency always benefits the system remains unclear. We conducted a two-step experiment (Fig. 1) to address this. First, *E. coli* was adapted to necromass recycling for 60 days in two contrasting matter-closed monocultures without nutrients: a heterogeneous environment (HTG) with quartz sand and a homogeneous liquid environment (HMG). Subsequently, the adapted *E. coli* populations and their wild-type (WT) progenitor were co-cultured with *C. reinhardtii* in spatially structured microbial biospheres to assess microbial population dynamics and system persistence. We hypothesized that adapted *E. coli* exhibit enhanced necromass recycling compared to the WT progenitor, thereby increasing biosphere stability and persistence during co-cultivation.

## Results
### Changes in *E. coli* phenotypes following monoculture
After 60 days of monoculture in a closed system, we recovered *E. coli* populations from both HTG and HMG spatial contexts (Fig. 1). The cell concentration in the HMG (median = 6.11, $\log_{10}$ CFU $mL^{-1}$) was numerically higher than that in the HTG (median = 1.70, $\log_{10}$ CFU $mL^{-1}$); although this difference was not statistically significant (Fig. 2A). To ensure reproducible and comparable phenotypic assessments, all recovered *E. coli* populations were subjected to a standardized preconditioning culture procedure prior to analysis. Briefly, this involved inoculation from glycerol stocks into Luria-Bertani (LB) medium for overnight growth, followed by a second overnight passage, ensuring approximately 20 generations of growth in a consistent physiological state before testing.

Following this conditioning, we observed substantial phenotypic adaptations in the pre-adapted populations compared to the WT progenitor

(Table 1). Specifically, qualitative assessments showed that two-thirds of the pre-adapted populations, comprising all HMG-derived populations (HMG1-3) and one HTG-derived population (HTG3), exhibited reduced motility[26], loss of curli fimbriae expression[27], and enhanced biofilm formation[28].

Biofilm formation is a key survival strategy under resource limitations, as it promotes nutrient retention and stress resistance and is often traded off against motility[29]. We therefore focused our quantitative analysis on this trait to assess its temporal dynamics and potential link to spatial structure during adaptation. Quantitative crystal violet assays[26] for biofilm formation revealed a clear temporal trend, with biofilm abundance increasing over time, where the HMG-derived populations consistently formed the most biofilm and WT progenitor the least across all three incubation periods (Supplementary Fig. 1). After 48 h, the biofilm formation ability of each population peaked, with HMG-derived populations significantly exceeding the WT progenitor but not differing significantly from HTG-derived populations (Fig. 2B). Furthermore, pellicle-formation tests[30] revealed that half of the adapted populations (HTG1, HTG3, HMG2) did not form pellicles (Table 1).

These results indicate that spatial structure strongly shaped microbial responses to resource-limited environments. This influence was evident in both bacterial abundance and phenotypic changes, with changes being more pronounced in HMG than in HTG.

### Enhanced carbon source utilization in *E. coli* post-monoculture
We determined growth of WT progenitor and derived pre-adapted populations across eight distinct carbon sources. Growth was analyzed by fitting logistic growth models, from which three key parameters were extracted: maximum growth rate (μ, reflecting substrate utilization velocity), carrying capacity (K, indicating substrate conversion efficiency), and area under the curve (AUC, representing comprehensive substrate exploitation capacity)[31]. Here, 'carrying capacity' refers to the maximum load in batch conditions before resource depletion. For the two insoluble necromass substrates, growth was monitored manually; in addition to μ, K, and AUC, we also calculated the doubling time as a specific measure of growth initiation rate (see "Methods").

Pre-adapted populations exhibited consistent growth pattern alterations compared to their WT progenitor across the six carbon sources tested (LB, M9+glucose, M9+gluconate, M9+glycerol, M9+xylose, and M9+acetate). Specifically, maximum growth rates decreased, whereas both carrying capacity and AUC increased (Supplementary Figs. 2-3). The

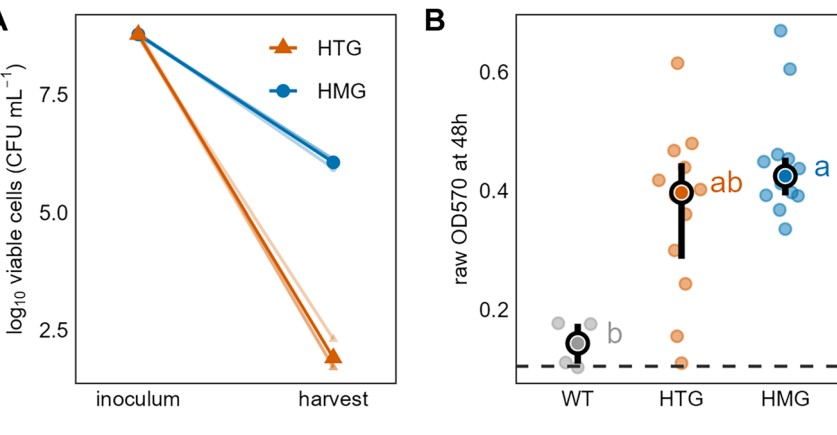

**Fig. 2 | Changes in cell counts and biofilm formation capacity of *Escherichia coli* after monoculture. A** Log₁₀-transformed viable cell counts of *E. coli* at the initiation and upon destructive harvest after 60 days of monoculture in two distinct systems (HTG: spatially heterogeneous; HMG: spatially homogeneous). Thin semi-transparent lines and small points represent the three biological replicates per treatment ($n = 3$). Thick lines and large points indicate the group means. While the mean cell concentration in the HMG at harvest was numerically higher than in the HTG, the difference was not statistically significant (Mann-Whitney U test, $W = 9$, $p = 0.077$). **B** Comparison of biofilm formation capacity among treatment groups at 48 h (WT: $n = 4$; HTG/HMG: $n = 12$). Data points represent OD570 values from the quantitative tissue culture plate (TCP) assay. The dashed line indicates the OD570 value of the negative control. Vertical error bars denote the interquartile range; white-circled points represent medians. Means without a common letter differ significantly (Dunn's test, $\chi^2(2) = 9.11$, $\eta^2 = 0.284$, $p = 0.008$ for HMG vs. WT).

HMG-derived populations showed the most pronounced changes, with significantly elevated AUC on all carbon sources except xylose (Fig. 3A, Dunn's test, $p < 0.05$). Furthermore, when grown on necromass from *C. reinhardtii* (M9+CreiNecro) or *E. coli* (M9+EcoNecro) as sole carbon sources, the pre-adapted populations demonstrated shortened doubling times and increased AUC relative to their WT progenitor (Supplementary Fig. 4). The divergence was again more prominent in HMG-derived populations than in HTG-derived populations, with HMG-derived populations exhibiting significantly higher AUC on both necromass-derived carbon sources (Dunn's test, $p < 0.05$).

Using AUC as a composite metric, we found that adaptation in the closed system increased the carbon utilization capacity of *E. coli*, and this increase was strongly dependent on spatial structure during adaptation. HMG-derived populations exhibited a significantly higher resource utilization capacity than the WT progenitor on seven of the eight tested carbon sources (all except xylose) (Fig. 3B; statistical details are provided in Supplementary Figs. 2–4). While HTG-derived populations also showed increased capacity relative to WT progenitor on most carbon sources (excluding xylose), the magnitude of increase was consistently less pronounced than that of the HMG-derived populations. Overall, the enhanced carbon utilization capacity (higher AUC) generally correlated positively with carrying capacity, particularly in adapted populations versus their WT progenitor (Fig. 3C and Supplementary Fig. 5). On *E. coli*-derived necromass (M9+EcoNecro), carbon utilization capacity correlated positively with maximum growth rate in HTG-derived populations, and with carrying capacity in HMG-derived populations.

## Co-culture of *C. reinhardtii* and *E. coli* in microbial biospheres

Pre-adapted *E. coli* populations or their WT progenitor were co-cultured with *C. reinhardtii* in spatially structured microbial biospheres. Photosynthetic activity, monitored via chlorophyll fluorescence, served as a proxy for biosphere persistence (Fig. 1). All co-cultures ceased photosynthetic activity within 10 days, indicating system-level failure (Supplementary Fig. 6). Using generalized additive models (GAMs) to determine the time to chlorophyll fluorescence loss (TCFL), we found that WT progenitor-containing biospheres persisted significantly longer (6.27 ± 1.00 days) than most pre-adapted populations (4.87 ± 0.55 days), except for HTG1 (5.74 ± 0.46 days; Fig. 4A).

Destructive sampling at days 10, 16, 23, and 28 showed that individual species (i.e., *E. coli* or *C. reinhardtii*) could still persist after system-level

failure. *E. coli* populations in WT biospheres maintained the highest cell densities at all time points (Fig. 4B). By contrast, the HMG-derived populations showed significantly reduced viability, falling below HTG-derived populations after day 23 (Supplementary Fig. 7). *C. reinhardtii* regrowth assays showed sporadic survival over time and across replicates, with marginally higher persistence in HMG biospheres (Fig. 4C). Crucially, *E. coli* extinction occurred only in the pre-adapted populations (not the WT), with HMG biospheres showing increasing extinction frequency, culminating in no detectable viable cells in HMG2/HMG3 by day 28 (Supplementary Fig. 8).

After 28-days of co-cultivation, surviving *E. coli* populations continued to diverge in motility and biofilm formation. The initially motile HTG-derived population HTG2 completely lost motility while increasing biofilm formation (Fig. 4D). By contrast, HTG1 retained minimal motility. Notably, the WT progenitor lost motility and exhibited biofilm formation within the 28-day period. Biofilm production increased temporally in all populations except HTG1, consistent with post-monoculture trends. Furthermore, all co-cultured populations, excluding HTG1, converged toward intermediate biofilm formation ability (Fig. 4E), suggesting a consistent trajectory of the phenotype.

Thus, in resource-limited microbial biospheres, *E. coli* progressively abandoned motility while increasing and converging toward intermediate biofilm production. Counterintuitively, the increased resource utilization capacity acquired during monoculture failed to improve co-culture persistence. Moreover, despite their enhanced necromass exploitation ability, these pre-adapted populations failed to sustain high population densities in co-culture. These results underscore how prior evolution under monoculture resource constraints critically shaped coculture outcomes.

## Factors influencing the persistence of co-cultured microbial biospheres

We employed survival analysis using the Kaplan-Meier estimator[32] to assess how *E. coli* phenotypes influenced the persistence of co-cultured microbial biospheres. Biospheres containing *E. coli* populations with higher necromass utilization capacity (higher AUC) and higher growth rates exhibited significantly lower biosphere persistence than those with lower AUC and growth rates (Supplementary Fig. 9A). Notably, resource utilization efficiency exerted divergent effects: biospheres with *E. coli* populations proficient in utilizing *E. coli*-derived necromass (higher carrying capacity) showed enhanced persistence. Conversely, proficiency in utilizing

**Table 1 | Phenotypic test results for the wild-type and evolved populations of *Escherichia coli* MC4100**

| Sample | Treatment[a] | Motility[b] | Curli Fimbriae[b] | Biofilm Forming (CRA[c]) | Biofilm Forming (TCP-24[d]) | Biofilm Forming (TCP-48[d]) | Biofilm Forming (TCP-72[d]) | Pellicle Forming[e] |
|---|---|---|---|---|---|---|---|---|
| WT | WT | Positive | Positive | Negative | None | Weak | Weak | Weak |
| HTG1 | HTG | Positive | Positive | Negative | None | Moderate | Weak | None |
| HTG2 | | Positive | Positive | Negative | Moderate | Strong | Moderate | Moderate |
| HTG3 | | Negative | Negative | Positive | Moderate | Moderate | Strong | None |
| HMG1 | HMG | Negative | Negative | Positive | Moderate | Moderate | Moderate | Weak |
| HMG2 | | Negative | Negative | Positive | Moderate | Strong | Strong | None |
| HMG3 | | Negative | Negative | Positive | Strong | Strong | Moderate | Weak |

[a]The experiment comprised three treatments: the untreated wild-type progenitor (WT), and strains subjected to monoculture in two distinct enclosed systems (HTG: spatially heterogeneous; HMG: spatially homogeneous).
[b]Motility and curli fimbriae expression were assessed qualitatively using motility test media and Congo Red indicator (CRI) agar, respectively, with results presented on a binary scale[26].
[c]CRA denotes the qualitative assessment of biofilm formation using Congo Red agar, with results also on a binary scale[28].
[d]TCP refers to the quantitative measurement of biofilm formation using the tissue culture plate (TCP) method; the accompanying numbers indicate the three measurement time points at 24, 48, and 72 hours. The biofilm-forming capacity of each population was categorized by comparing the raw OD570 values with those of the negative control[28].
[e]Pellicle formation was similarly evaluated qualitatively, following the TCP protocol but with cultures grown in test tubes[30].

*C. reinhardtii*-derived necromass led to reduced system persistence. Further, motile *E. coli* expressing curli fimbriae have significantly increased biosphere persistence, whereas moderate-to-strong biofilm formation substantially decreased biosphere persistence (Fig. 5A, pairwise log-rank test, $p < 0.001$). Cox proportional hazards modeling[33] confirmed these trends: compared to strains with low motility, higher bacterial motility was associated with a reduced risk of system failure by 77% (Fig. 5B and Supplementary Fig. 9B; HR = 0.23, $p < 0.01$). Conversely, compared to populations with weak pellicle-forming ability, moderate pellicle-forming ability increased this risk by 161% (HR = 2.61, $p < 0.01$).

Spearman correlation analysis revealed significantly negative associations between biofilm formation and motility, but strong positive correlations between biofilm formation and carbon sources utilization capacity (Supplementary Fig. 10). A shorter time to chlorophyll fluorescence loss correlated strongly with increased biofilm formation and higher carbon source utilization capacity. Conversely, longer time to chlorophyll fluorescence loss was associated with improved motility by *E. coli* (Fig. 5C).

Our results demonstrate a trade-off between motility and biofilm formation in *E. coli* populations with respect to resource acquisition. Crucially, enhanced resource utilization capacity by *E. coli* consistently correlated negatively with co-cultured system persistence—evidenced by both lower survival probability and shorter time to chlorophyll fluorescence loss.

## Discussion

We investigated how resource limitation in spatially structured environments shapes survival strategies using a microbial biosphere model. *E. coli* sustained viable populations for 60 days in matter-closed systems via necromass recycling, regardless of spatial homogeneity[13]. These pre-adapted populations underwent phenotypic shifts (Fig. 6A) that significantly altered subsequent co-culture dynamics with *C. reinhardtii*.

### Evolutionary trade-offs between individual adaptation and system persistence

Nutrient depletion below critical thresholds typically results in ecosystem collapse in microbial biospheres[22]. We hypothesized that efficient necromass recycling by the decomposer would prolong system persistence by expanding the cycling nutrient pool. However, phenotypic shifts acquired during resource-limited monoculture unexpectedly accelerated collapse in subsequent co-cultures (Fig. 6B). This outcome may be explained by several non-mutually exclusive mechanisms: First, intensified resource competition likely played a key role. While *E. coli* and *C. reinhardtii* engage in reciprocal material exchange ($CO_2$/inorganic nutrients for organic carbon)[34,35], *C. reinhardtii* is a facultative heterotroph that competes for organic substrates[25]. Pre-adapted *E. coli* (particularly HMG-derived populations) exhibited enhanced carbon utilization, potentially outcompeting *C. reinhardtii* for necromass-derived compounds. Although starved *E. coli* incompletely metabolizes endogenous compounds and necromass[36], and may resort to fermentation under oxygen-limited conditions, potentially releasing metabolites such as aromatic amino acids (e.g., phenylalanine) that could be utilized by *C. reinhardtii*[37], its aggressive resource acquisition likely overwhelms this potential benefit and may suppress *C. reinhardtii*.

Second, the loss of motility in pre-adapted *E. coli* likely disrupts spatial coupling with the producers. Bacterial motility facilitates proximity to *C. reinhardtii*, enabling efficient aerobic respiration and nutrient exchange[38,39]. While motile populations (WT progenitor and HTG-derived populations HTG1/HTG2) could forage for *C. reinhardtii* exudates, HMG-derived populations lost motility and increased biofilm formation. This sessile lifestyle likely sequesters nutrients within the biofilm matrix. Since diffusion is a slow process, this sequestration privatizes and hoards shared nutrient resources (the "commons") for immediate local benefit, while weakening interaction strength[40]. This preferential resource retention limits reciprocal nutrient exchange with *C. reinhardtii*. Since algal proliferation depends on the accumulation of $CO_2$ and recycled nutrients[41], this limitation reduces algal photosynthetic activity and potentially depletes oxygen availability.

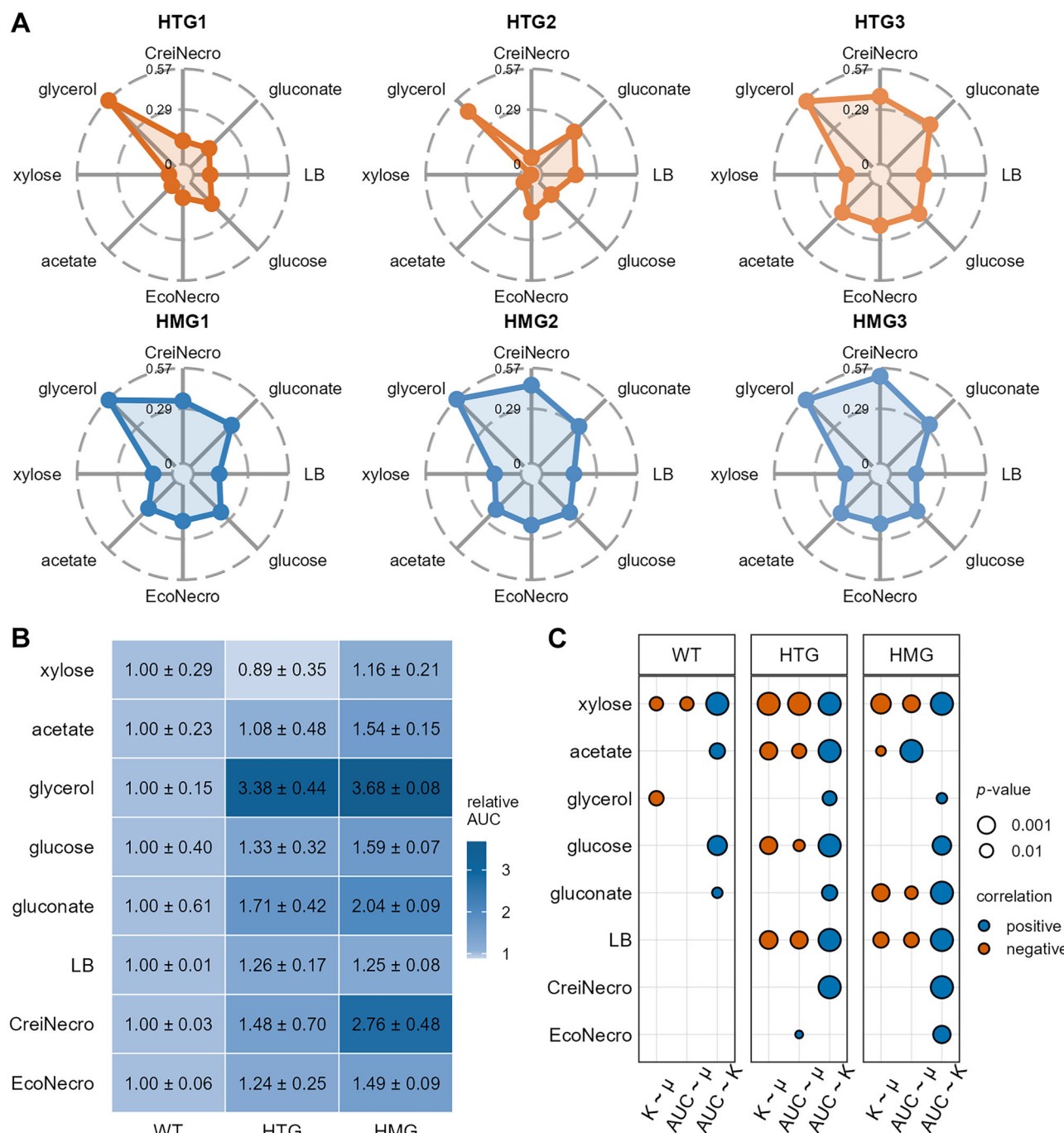

**Fig. 3 | Enhanced carbon source utilization capacity of *Escherichia coli* after monoculture adaptation. A** Comparison of the area under the curve (AUC) for growth curves on eight carbon sources between the wild-type progenitor (WT) and three evolved populations from each monoculture system (HTG: spatially heterogeneous; HMG: spatially homogeneous). Data were normalized to the AUC of the WT and $\log_{10}$-transformed. Consequently, the AUC for the WT is 0 for each carbon source; values > 0 indicate an increase relative to WT, while values < 0 indicate a decrease. CreiNecro and EcoNecro denote the necromass of *Chlamydomonas reinhardtii* and *E. coli*, respectively. The original AUC data are presented as beeswarm plots in Supplementary Figs. 2-4. **B** Heatmap comparing the AUC for growth curves on eight carbon sources among the three treatment groups (WT, HTG, HMG). Data were similarly normalized to the WT AUC but without $\log_{10}$ transformation. Values are displayed as mean ± standard error (SE). **C** Correlations among three growth parameters, maximum growth rate (μ), carrying capacity (K), and AUC, for the three treatments across different carbon sources. The original data are presented as scatter plots in Supplementary Fig. 5. Correlations were assessed using ordinary least squares (OLS) regression; only statistically significant results are shown. Blue and orange indicate positive and negative correlations, respectively. Point size represents the absolute value of the *p*-value, with larger *p*oints corresponding to smaller *p*-values. For (**A**), n = 3 for CreiNecro and EcoNecro; n = 4 for gluconate, glucose, glycerol, and LB; and n = 6 for acetate and xylose. **B**, **C** pool the three HTG and three HMG populations into their respective treatment groups for comparison with WT.

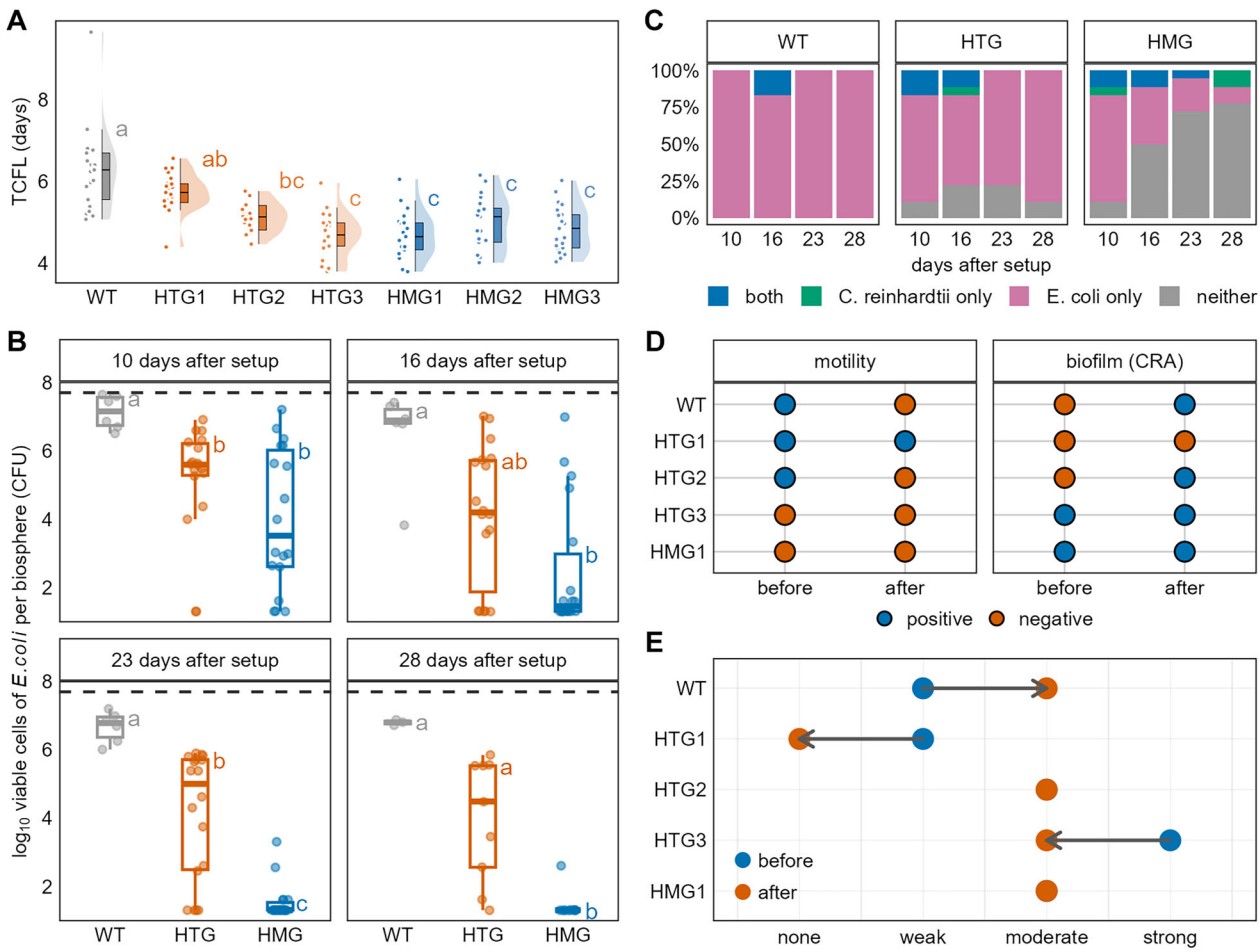

**Fig. 4 | Persistence and growth dynamics of the co-cultured microbial biospheres.**
**A** Time to chlorophyll fluorescence loss (TCFL) for microbial biospheres co-cultured with the wild-type progenitor (WT) and the six evolved *Escherichia coli* populations ($n = 21$ for each population). TCFL represents the persistence of the co-culture system. Orange hues denote populations pre-cultured in the HTG (spatially heterogeneous) system; blue hues denote those from the HMG (spatially homogeneous) system. Means without a common letter differ significantly (Dunn's test, $\chi^2(6) = 66.994$, $\eta^2 = 0.436$, $p < 0.01$). **B** $Log_{10}$-transformed *E. coli* population densities (cells per biosphere) for the three treatments (WT, HTG, and HMG) at the four destructive harvest time points, with $n = 6$ (WT) a*n*d 18 (HTG/HMG) at days 10, 16, and 23, and $n = 3$ (WT) a*n*d 9 (HTG/HMG) at day 28. At each time point, means without a common letter differ significantly among treatments (Dunn's test: Day 10, $\chi^2(6) = 14.286$, $\eta^2 = 0.315$, $p < 0.05$; Day 16, $\chi^2(2) = 13.104$, $\eta^2 = 0.285$, $p = 0.001$; Day 23, $\chi^2(2) = 26.243$, $\eta^2 = 0.622$, $p < 0.05$; Day 28, $\chi^2(6) = 15.022$, $\eta^2 = 0.723$, $p < 0.05$). The dashed line at the top indicates the initial number of *E. coli* cells inoculated per biosphere. **C** The percentage of biospheres in each state relative to the total number

harvested per time point for the three treatments across the four harvests (*n* values as in **B**). "both" indicates biospheres where both *E. coli* and *Chlamydomonas reinhardtii* survived; "neither" indicates the opposite. "*C. reinhardtii* only" and "*E. coli* only" indicate biospheres where only the alga or the bacterium survived, respectively. **D** Comparison of motility and biofilm formation capacity before and after co-culture for the five *E. coli* populations successfully recovered at the fourth harvest. Motility and biofilm formation were assessed qualitatively using motility test media and Congo Red agar (CRA), respectively, with results presented on a binary scale[26,28]. **E** Classification of biofilm formation capacity for the five recovered populations before and after co-culture. Biofilm formation was quantified using the tissue culture plate (TCP) method (72-hour data). Populations were categorized based on a comparison of the measured raw OD570 values with those of the negative control. Blue and orange points represent biofilm-forming capacity before and after co-culture, respectively; arrows indicate the direction of change.

Consequently, due to this lack of oxygen, *E. coli* may be forced into less efficient anaerobic metabolism, relying primarily on the fermentation of necromass. These combined effects create a "tragedy of the commons" (wherein individual optimization undermines collective stability) and likely impede coexistence.

Third, system collapse may involve inhibitory metabolite accumulation. The rapid decline in photosynthetic activity suggests *C. reinhardtii* suppression, which could also be driven by incomplete necromass degradation or the buildup of toxic metabolic products[42]. Such metabolic by-products could directly inhibit algal physiology, contributing to system failure.

Consequently, collapse likely stems from a synergy of resource competition, disrupted coupling, and metabolic inhibition rather than a single resource depletion.

Finally, evolutionary mismatches between monoculture adaptation and co-culture conditions may have accelerated collapse. In small, fragmented populations, genetic drift can fix deleterious mutations or cause the loss of beneficial alleles[43]. Traits that are beneficial in monoculture can become detrimental in co-culture, as observed in shifting selection pressures in stationary-phase batch cultures[16]. The synergy of drift-driven maladaptation and rapid resource exhaustion likely contributed to population extinction[44].

**Environmental drivers of phenotypic evolution in monocultures**
The repeatable evolution of *E. coli* observed after preadaptation in resource-limitation indicates strong selective pressures prioritizing starvation survival over growth rate[12], as evidenced by increased population carrying capacities. In homogeneous systems (HMG), the high energetic cost of motility likely

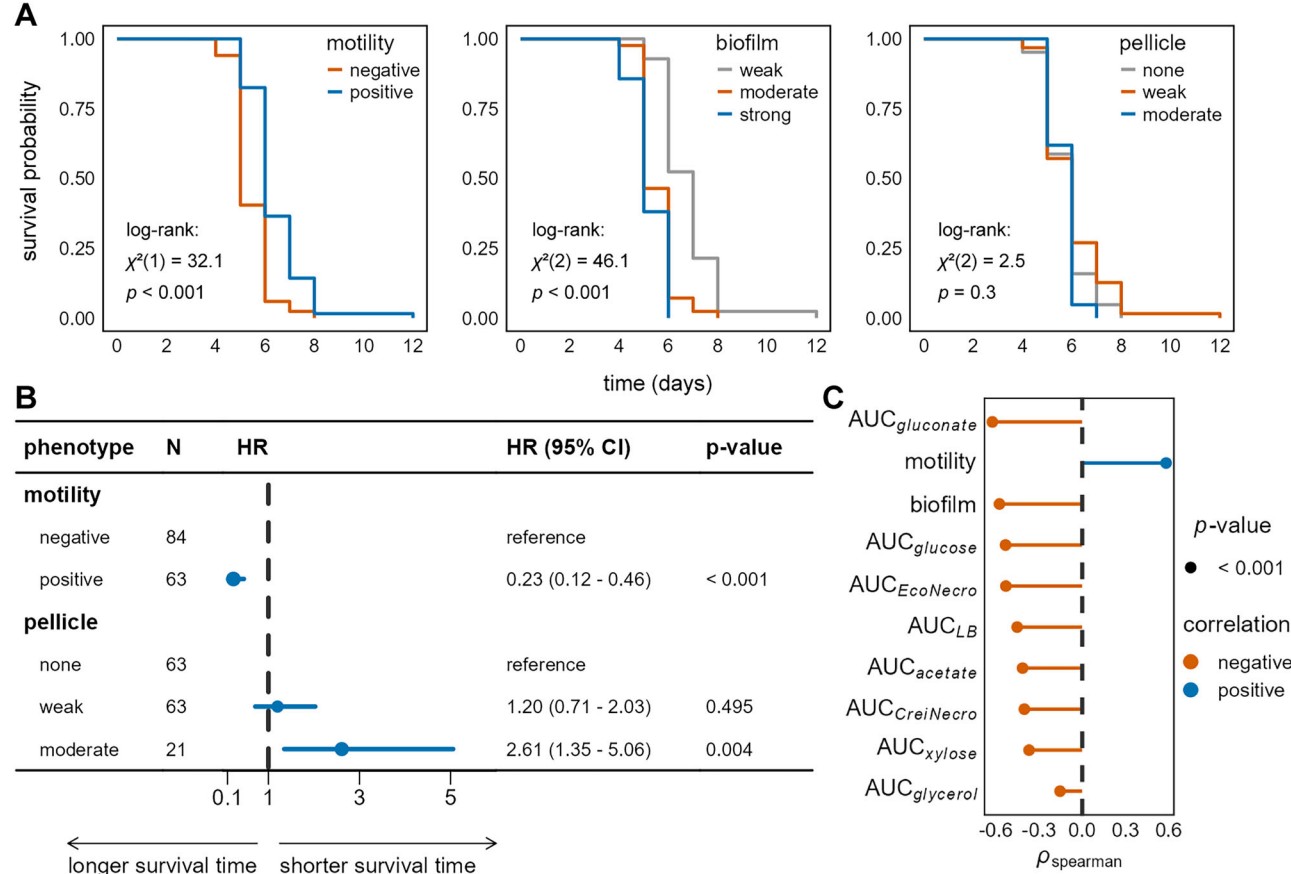

**Fig. 5 | Factors influencing the persistence of microbial biospheres. A** Kaplan-Meier survival probability curves for microbial biospheres, grouped by four *Escherichia coli* phenotypic traits. Motility was assessed qualitatively using motility test media with results on a binary scale[26]. Biofilm formation capacity was quantified using the tissue culture plate (TCP) method (72 h data)[26]. Populations were categorized based on comparing the raw OD570 values with those of the negative control. Pellicle formation was similarly assessed qualitatively, following the TCP protocol but with cultures grown in test tubes[30]. The number of biospheres (*n*) in each phenotypic group is as follows: motility (positive, *n* = 63; negative, *n* = 84); biofilm (strong, *n* = 21; moderate, *n* = 84; weak, *n* = 42), and pellicle (moderate, *n* = 21; weak, *n* = 63; none, *n* = 63). **B** Forest plot presenting the results of the Cox proportional hazards model. HR = hazard ratio; CI = confidence interval; *N* = sample size. **C** Spearman's rank correlation coefficients and statistical analysis results between the persistence of co-cultured microbial biospheres (characterized by time to chlorophyll fluorescence loss) and the following traits of the *E. coli* populations: the AUC of growth curves on eight carbon sources, motility, and biofilm formation capacity. Biofilm data for correlation analysis were raw OD570 values from the quantitative TCP assay (72 h), not classified categories. The complete correlation heatmap is shown in Supplementary Fig. 10. Orange and blue represent negative and positive correlations, respectively. Point size corresponds to *p*-values; all *p*-values were < 0.001.

**Fig. 6 | Conceptual diagram showing *Escherichia coli* adaptations under nutrient limitation and its impact on community persistence. A** Proposed changes in motility, biofilm formation capacity, and carbon source utilization of *E. coli* under nutrient-limited conditions in closed monoculture systems and in microbial biospheres. Shaded areas indicate inferred trends. **B** Hypothesized and experimentally inferred relationship between *E. coli* carbon source utilization capacity and the persistence of co-cultured microbial biospheres.

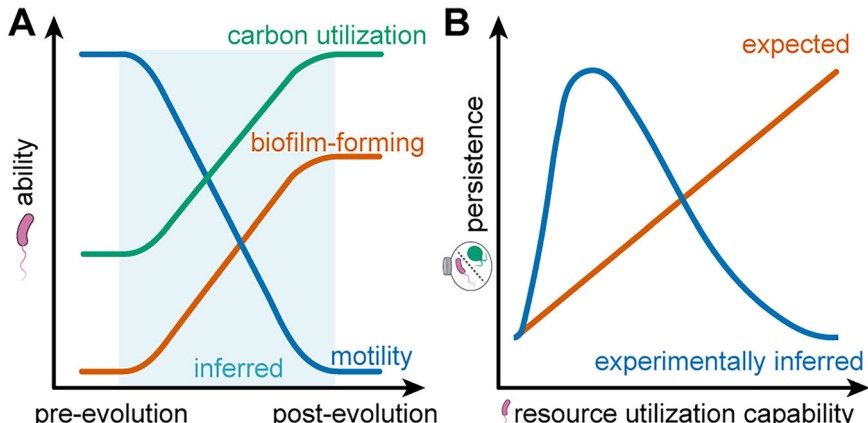

drove its ubiquitous loss[45]. Conversely, while motility enables resource exploration in heterogeneous systems (HTG), its maintenance becomes unsustainable under chronic starvation[46,47]. The further acceleration of motility loss in the WT progenitor during co-culture suggests that species interactions can accelerate metabolic evolution[48]. Notably, the partial motility retention in HTG1 represents a behavioral legacy that likely enhanced its foraging efficiency and persistence in co-cultures.

Resource limitation also favors biofilm formation to improve resource capture and stress resistance[29], creating a distinct trade-off with motility. Biofilm formation strongly correlated with expanded carbon utilization capacity in pre-adapted *E. coli* populations. While spatial structure (HTG) promoted phenotypic heterogeneity and niche-specific specialization[49], HMG conditions favored phenotypic convergence and broader carbon utilization. Spatial fragmentation in HTG potentially restricts adaptation by limiting access to necromass[21,50]. By contrast, well-mixed HMG environments facilitate metabolic diversification and resource partitioning, which may enhance population persistence by mitigating negative cellular interactions[51].

Although genomic mutations were not sequenced, the persistent and directional phenotypic shifts observed likely reflect stable genetic adaptations. In microbial experimental evolution, adaptive genetic changes arise when selective pressures drive differential survival[43]. Rapid adaptation under nutrient restriction, including the emergence of GASP phenotypes, is well-documented within comparable timescales[16,18,52]. The maintenance of distinct phenotypic profiles through approximately 20 generations of recovery and cultivation in nutrient-rich media further supports a genetic basis, as this process would dilute transient epigenetic effects[53,54]. The lack of reversion to wild-type phenotypes and the continued directional changes during co-culture confirm that these observed traits represent stable evolutionary responses to their respective environmental conditions.

### Limitations and future perspectives

This study has several limitations. First, the approximately 20-generation incubation in LB medium required for diagnostic phenotyping introduced different selective pressures compared to the original closed-system environments (HTG and HMG). We emphasize that this standardized, nutrient-rich assay was designed as a controlled diagnostic tool to quantify heritable fitness changes acquired during long-term selection, not to mimic the co-culture environment. Second, while limited replication ($n = 3$ per treatment) restricts inferences on among-lineage variability, the robust convergence of traits, such as motility loss in HMG-derived populations, underscores the strength of the observed selection. Third, motility was assessed using a qualitative binary method, which may not capture the full continuum of motility phenotypes. Finally, our results link specific traits to time to chlorophyll fluorescence loss correlatively; establishing causality requires targeted manipulative experiments.

Biosphere collapse likely stems from intertwined eco-evolutionary mechanisms, including intensified competition, disrupted motility-driven coupling, and potential metabolic dysregulation, rather than a single cause. Future research should (i) employ genomics to link the observed phenotypic convergences to specific genetic mutations; (ii) conduct in situ assays to verify phenotypes under authentic biosphere conditions; (iii) manipulate specific traits, for instance by utilizing *E. coli* mutants deficient in motility or enhanced in biofilm production, to test causal links with system persistence; and (iv) monitor environmental variables (e.g., $O_2$, pH, metabolites) to differentiate resource exhaustion from toxic accumulation. Additionally, investigating whether these evolutionary trajectories, particularly the trade-off between motility and biofilm formation, emerge similarly within biospheres containing diverse facultative aerobic bacteria and multispecies communities will clarify how community-context evolution shapes ecosystem persistence.

In conclusion, resource-limited adaptation can yield counterintuitive outcomes: traits enhancing decomposer performance in isolation may diminish producer-decomposer system persistence during interspecific interactions. Spatial structure during pre-adaptation modulates these

evolutionary trajectories, thereby altering subsequent community dynamics. By manipulating evolutionary history prior to assembly, we provide a framework to predict and test evolutionary influences on ecosystem-level functions. Future mechanistic studies are needed to resolve the causes of collapse and generalize these findings across diverse taxa and environmental conditions.

## Methods

### Bacterial strain, media, and monoculture

*E. coli* MC4100 was obtained from the Leibniz-Institut DSMZ (DSM 6574). Cells were revived by streaking frozen stocks onto LB agar plates (10 g L$^{-1}$ tryptone, 5 g L$^{-1}$ yeast extract, 5 g L$^{-1}$ NaCl, 1 mL L$^{-1}$ 1 N NaOH, 15 g L$^{-1}$ agar) and incubating them overnight at 37 °C. A single colony was inoculated into LB medium (10 g L$^{-1}$ tryptone, 5 g L$^{-1}$ yeast extract, 5 g L$^{-1}$ NaCl, 1 mL L$^{-1}$ 1 N NaOH) and grown under shaking conditions at 150 rpm and 37 °C for 18 h. After incubation, 1 mL aliquots were serially diluted in saline for cell quantification using the SP-SDS method[55]. Glycerol stocks were prepared from overnight cultures by mixing *E. coli* cells with a 30% glycerol solution (1:1 ratio) and storing at -70°C, serving as the WT progenitor stock. Remaining cultures were centrifuged at 4000 rpm for 10 mins, washed twice with saline, and resuspended for further experiments.

For monoculture adaptation, matter-closed systems were incubated in a climate chamber at 25°C with a 16-hour light/8-hour dark cycle and 19.4% humidity for two months. Two configurations were used:

Heterogeneous group (HTG): 600 mL of 1–2 mm quartz sand and 240 mL of *E. coli* suspension in static conditions.

Homogeneous group (HMG): 280 mL of *E. coli* suspension incubated with constant shaking at 150 rpm.

For each configuration, three biological replicate systems were established, incubated in 1 L glass bottles sealed with Plastic-Fermit (Fermit GmbH, Germany; Supplementary Fig. 11A). Cells from HTG systems were recovered by washing quartz sand with 240 mL of saline. Harvested samples were used to prepare glycerol stocks by incubating 1 mL aliquots in LB media overnight. These pooled cultures provided comprehensive representations of the monocultured populations.

For subsequent phenotypic and growth assays, frozen glycerol stocks were recovered and used to inoculate a preconditioning culture: a single overnight culture in LB medium. This step ensured consistent and active starting cultures and acclimated cells to standard laboratory conditions after long-term incubation in saline. Following this, a fresh overnight culture was initiated from the preconditioning culture to generate the cells used directly in testing. Thus, between recovery from the closed monoculture system and final assessments, cells underwent approximately 20 generations of growth in rich, homogeneous LB medium.

### Algal strain and growth conditions

*C. reinhardtii* strain 11-32b was obtained from the Culture Collection of Algae (SAG) at Göttingen University. Cells were maintained on Tris-Acetate-Phosphate (TAP) slants (20 mL L$^{-1}$ 1 M Tris base, 1 mL L$^{-1}$ Phosphate buffer II, 10 mL L$^{-1}$ Solution A, 1 mL L$^{-1}$ Hutner's trace element solution, 1 mL L$^{-1}$ CH$_3$COOH, 15 g L$^{-1}$ agar) and transferred to Tris-Acetate-Phosphate (TAP) medium (20 mL L$^{-1}$ 1 M Tris base, 1 mL L$^{-1}$ Phosphate buffer II, 10 mL L$^{-1}$ Solution A, 1 mL L$^{-1}$ Hutner's trace element solution, 1 mL L$^{-1}$ CH$_3$COOH) as needed. Cultures were incubated at 25°C with shaking at 150 rpm for five days until reaching the stationary phase. Cell concentrations were determined using a Neubauer hemocytometer (Hecht Glaswarenfabrik GmbH, Germany) and a phase-contrast microscope (DM2500, Leica Mikrosysteme Vertrieb GmbH, Germany).

### Construction of standard curves and determination of cellular carbon content

Standard curves were constructed to correlate OD600 measurements with colony-forming units (CFU mL$^{-1}$). Glycerol-frozen stocks of *E. coli* were inoculated into LB media, incubated, and serially diluted. OD600 was measured in 96-well plates, and CFU counts were obtained using the SP-

SDS method[55]. Data were linearly fitted to generate regression equations, facilitating precise cell concentration calculations during experiments (Supplementary Figs. 12-13).

To determine cellular carbon content, *C. reinhardtii* and *E. coli* were grown to stationary phase, harvested by centrifugation, and freeze-dried. Three replicates were prepared for each sample. Dried samples were analyzed using a Euro EA-CHNSO Elemental Analyzer (HEKAtech GmbH, Germany) to determine carbon content per cell:

A single *C. reinhardtii* cell weighed ~$5.307 \times 10^{-8}$ mg, with a carbon content of $2.325 \times 10^{-8}$ mg.

A single *E. coli* MC4100 cell weighed ~$2.002 \times 10^{-9}$ mg, with a carbon content of $7.968 \times 10^{-10}$ mg.

## Utilization of carbon sources

The carbon utilization capacity of *E. coli* was assessed using LB medium and five defined carbon sources: glucose, glycerol, acetate, xylose, and gluconate. These six chemically distinct carbon sources tested divergent transport and metabolic pathways[56]. For instance, glucose is metabolized via glycolysis[57], while xylose is degraded using pentose phosphate pathway intermediate xylulose-5-phosphate before integration into glycolytic intermediates[58]. Each carbon source, except for LB medium, was added to M9 minimal medium (M9) (6 g L$^{-1}$ Na$_2$HPO$_4$, 3 g L$^{-1}$ KH$_2$PO$_4$, 1 g L$^{-1}$ NH$_4$Cl, 0.5 g L$^{-1}$ NaCl, 0.5 mg L$^{-1}$ FeSO$_4$·7H$_2$O, 1 mL 1 M L$^{-1}$ MgSO$_4$, 1 mL L$^{-1}$ 0.1 M CaCl$_2$, 0.036 mg L$^{-1}$ (NH$_4$)$_6$Mo$_7$O$_{24}$·4H$_2$O, 0.248 mg L$^{-1}$ H$_3$BO$_3$, 0.072 mg L$^{-1}$ CoCl$_2$·6H$_2$O, 0.024 mg L$^{-1}$ CuSO$_4$·5H$_2$O, 0.16 mg L$^{-1}$ MnCl$_2$, 0.016 mg L$^{-1}$ ZnSO$_4$·H$_2$O) as the sole carbon source at a concentration standardized to 0.5 mol C L$^{-1}$. Cultures were inoculated into 96-well plates, incubated, and OD600 was measured at 10 min intervals using a FLUOstar Omega microplate reader (BMG Labtech Ltd., UK). For each sample, four replicate wells were set up for every carbon source, while six replicate wells were used for xylose and acetate. Carbon source solutions without added *E. coli* served as negative controls. Logistic growth curves were fitted using the R package *Growthcurver*[31] to calculate the maximum growth rate (μ), carrying capacity (K), and area under the curve (AUC). *Growthcurver* uses the logistic equation:

$$N_t = \frac{K}{1 + \left(\frac{K - N_0}{N_0}\right)e^{-rt}}$$

Where $N_0$ is the initial population size, $K$ is the carrying capacity possible in the given environment, and $r$ is the maximum growth rate that could occur in the absence of growth limitation.

## Necromass utilization

Since necromass is insoluble, its utilization by *E. coli* leads to a decrease in absorbance, and *E. coli* proliferation leads to an increase in absorbance. This counteracting effect makes automatic measurement of growth curves using a microplate reader impossible. To evaluate necromass utilization, *C. reinhardtii* and *E. coli* cultures were grown separately to the stationary phase, harvested, and autoclaved to create necromass solutions (0.05 mol C L$^{-1}$). *E. coli* was inoculated into necromass suspensions and incubated at 250 rpm and 37 °C. For each sample, three replicates were prepared for both necromass types. Necromass solutions without added *E. coli* served as negative controls. Cell concentrations were manually measured twice daily using the SP-SDS method[55], and doubling times were calculated as (Supplementary Figs. 14-15):

$$\text{Doubling Time} = \frac{\triangle Time}{\triangle Generations\,(n)} = \frac{t_2 - t_1}{\frac{\left(\log\left(\frac{b}{B}\right)\right)}{\log(2)}}$$

Where $t_1$ is timepoint 1, $t_2$ is timepoint 2, $b$ is the number of *E. coli* at $t_2$, $B$ is the number of *E. coli* at $t_1$, and $n$ is the number of generations.

Initial measurements began 17 h post-inoculation to account for lag phase interference, which may have slightly inflated doubling time estimates. Growth curves were analyzed using *Growthcurver*, consistent with the method for chemical carbon sources.

## Determination of *E. coli* phenotypes

To assess *E. coli* traits at the population level, we used standardized overnight cultures (generated via the preconditioning protocol described in 5.1), which represent a mixture of cells from each evolved population. These population-level cultures were diluted to concentrations of $1 \times 10^4$ CFU mL$^{-1}$ and $1 \times 10^6$ CFU mL$^{-1}$. These resulting suspensions were used to assess biofilm formation, pellicle formation, motility, and curli fimbriae expression.

Curli fimbriae expression. Curli fimbriae expression was assessed using Congo Red Indicator (CRI) plates (10 g L$^{-1}$ tryptone, 5 g L$^{-1}$ yeast extract, 5 g L$^{-1}$ NaCl, 1 mL L$^{-1}$ 1 N NaOH, 40 mg L$^{-1}$ Congo Red indicator, 15 g L$^{-1}$ agar, 20 mg L$^{-1}$ Brilliant Blue R). A 30 μL suspension of $1 \times 10^6$ CFU mL$^{-1}$ was pipetted onto the plates, which were incubated at 37°C, with results recorded at 24, 48, and 72 h. Each sample was tested in triplicate. CRI plates utilize a low sugar formulation, which minimizes interference from excessive, nonspecific polysaccharide production and specifically focuses on curli fimbriae expression. Colonies expressing curli fimbriae bind both the Congo Red and Brilliant Blue R dyes and exhibit a brown, dry, and rough (BDAR) morphology, while non-expressing colonies remain smooth and white (SAW)[27] (Supplementary Fig. 11B).

Biofilm formation. For qualitative assessment, $1 \times 10^6$ CFU mL$^{-1}$ cell suspensions were streaked onto Congo Red Agar (CRA) plates (0.8 g L$^{-1}$ Congo Red, 37 g L$^{-1}$ BHI, 36 g L$^{-1}$ C$_{12}$H$_{22}$O$_{11}$, 10 g L$^{-1}$ agar). The high concentration of sucrose in the medium was used to promote the production of extracellular polymeric substances. Plates were incubated at 37 °C, and results were recorded at 24 and 48 h. CRA visualizes extracellular polymeric substances characteristic of biofilms; biofilm formers typically produce black, dry, crystalline colonies, whereas non-formers produce red or pink colonies[28] (Supplementary Fig. 11C). Each sample was tested in triplicate. Quantitative analysis was performed using the tissue culture plate (TCP) method. 200 μL of cell suspensions ($1 \times 10^4$ CFU mL$^{-1}$) were added in quadruplicate to 96-well plates. Plates were incubated at 37 °C for 24, 48 and 72 h. Following incubation, OD600 was measured, and wells were aspirated, washed three times with VE-water, and air-dried. Biofilms were stained with 0.5% crystal violet (CV) solution (5.06 g L$^{-1}$ Crystal Violet, 253 mL L$^{-1}$ Methanol) for 20 min, followed by washing and ethanol extraction. Absorbance at 570 nm (OD570) was measured using a FLUOstar Omega microplate reader. Biofilm-forming ability was categorized based on OD570 values (ODS) compared to the negative control (ODC): ODS ≤ ODC = no biofilm producer, ODC < ODS ≤ (2 × ODC) = low biofilm producer, 2 × ODC < ODS ≤ 4 × ODC = moderate biofilm producer, and 4 × ODC < ODS = strong biofilm producer[26]. TCP data at 72 h were used as the primary indicator for biofilm production.

Motility testing. Motility was tested by inoculating $1 \times 10^6$ CFU mL$^{-1}$ cell suspensions into motility test media (10 g L$^{-1}$ tryptone, 5 g L$^{-1}$ NaCl, 5 g or 3 g L$^{-1}$ agar) using an inoculating needle to stab through the center of the media. Triplicate tests were set up for each sample. Plates were incubated at 37 °C and observed daily for seven days (Supplementary Fig. 11D). Non-motile samples show growth restricted to the inoculation line, while motile samples exhibit turbidity extending away from the inoculation line[26].

Pellicle formation. For pellicle formation, 100 μL of $1 \times 10^6$ CFU mL$^{-1}$ cell suspensions were inoculated into glass tubes containing 1.9 mL of LB and incubated statically at 25 °C for 5 days. Each sample was tested in triplicate. Pellicle formation at the gas-liquid interface was visually assessed and photographed (Supplementary Fig. 11E). Tubes were then washed with VE-water, air-dried, and stained with CV, and ethanol was used to extract the stain for OD570 measurement[30].

## Construction of co-cultured microbial biospheres

Co-cultured microbial biospheres containing *E. coli* and *C. reinhardtii* were prepared in the inner 24 wells of 48-well plates (GREINER GmbH,

Germany). Each well was filled with approximately 770 mg of quartz sand to provide structural support. Cultures of *C. reinhardtii* and *E. coli* in the stationary phase were harvested by centrifugation at 4000 rpm for 5 min, washed three times with saline, and resuspended in saline to final concentrations of $1 \times 10^7$ cells mL$^{-1}$ for *C. reinhardtii* and $5 \times 10^8$ cells mL$^{-1}$ for *E. coli*. Equal volumes of these suspensions were mixed (1:1 v/v), yielding a co-culture inoculum with an initial *C. reinhardtii* to *E. coli* cell count ratio of 1:50.

A total of 200 μL of the mixed cell suspension was added to each well to establish an independent microbial biosphere. Control wells received 200 μL of saline. Each plate included three biological replicate biospheres for each *E. coli* + *C. reinhardtii* sample, randomly distributed across the plate. A total of seven plates were prepared, resulting in 21 experimental biosphere replicates per sample and 21 negative control replicates across the entire experiment. Plates were sealed with ThermalSeal RTS sealing film (Excel Scientific, Inc., USA) to prevent material exchange and incubated statically in a climate chamber at 25 °C with a 16 h light/8 h dark cycle and 19.4% humidity.

Chlorophyll fluorescence was measured periodically using the FLUOstar Omega microplate reader with excitation and emission wavelengths of 470 nm and 680 nm, respectively[59]. Biospheres were destructively harvested on days 10, 16, 23, and 28, with harvested samples used to assess population viability. For the first three time points, two plates were harvested at each time point, providing 6 biosphere replicates per sample per harvest. At the final point, the remaining plate was harvested, providing 3 replicates per sample.

### Statistics and reproducibility
All statistical analyses were performed using R (v4.4.0)[60]. As preliminary diagnostics indicated violations of parametric test assumptions, non-parametric methods were employed throughout. Spearman's rank correlation was used to assess associations between phenotypic traits, carbon utilization capacities, and system persistence. For comparisons across three or more groups, the Kruskal-Wallis rank sum test was applied, with significant results followed by Dunn's test (*p*-values adjusted via the Bonferroni method). For comparisons between two groups, the Wilcoxon rank sum test (Mann-Whitney U test) was employed. The effect size of the test was expressed using $\eta^2$. Generalized additive models (GAMs) were used to process chlorophyll fluorescence time-series data, calculating the time to fluorescence loss for each biosphere. Survival analysis was conducted on chlorophyll fluorescence data to compare the persistence of co-cultured microbial biospheres. Kaplan-Meier estimators[32] were used to estimate survival functions, and the log-rank test was applied to compare groups. Pairwise comparisons between groups were conducted using the *pairwise_survdiff()* function from the *survminer* package, with Bonferroni correction for multiple comparisons. To analyze the influence of multiple variables, the semiparametric Cox PH model was used to model the survival data[33]. The model was optimized using the *stepAIC()* function from the *MASS* package based on the Akaike information criterion (AIC). The proportional hazards assumption was evaluated using the *cox.zph()* function from the *survival* package and Schoenfeld residual plots.

For monoculture adaptations, three biological replicate systems were established per treatment ($n = 3$ for HTG/HMG). For carbon source utilization assays, the number of replicates varied: glucose, glycerol, gluconate, and LB ($n = 4$ per sample), xylose and acetate ($n = 6$ per sample), and necromass substrates ($n = 3$ per sample). For phenotypic characterization, each assay was performed in triplicate (motility, curli fimbriae, biofilm formation via CRA plates, and pellicle formation) or quadruplicate (quantitative biofilm formation via the TCP method). For co-culture microbial biospheres, 21 biological replicate systems were established per *E. coli* population. These replicates were randomly distributed across the plate. Destructive harvests were performed at days 10, 16, 23, and 28, yielding 6 replicates per time point for the first three harvests and 3 replicates for the final harvest.

### Reporting summary
Further information on research design is available in the Nature Portfolio Reporting Summary linked to this article.

## Data availability
The raw data and tidy data that support the findings of this study are available in Figshare with the identifier https://doi.org/10.6084/m9.figshare.30933734[61].

## Code availability
The code developed in this study has been publicly released through Figshare (https://doi.org/10.6084/m9.figshare.30933815)[62].

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

## Acknowledgements

We gratefully acknowledge Anja Wulf, Sabine Buchert, and Maximilian Hähnlein for their essential contributions to the experiments and insightful discussions.

## Author contributions

S.J.: Conceptualization, Investigation, Formal analysis, Visualization, Writing—Original draft; A.H.: Investigation, Formal analysis; J.A.: Visualization, Writing—Review & Editing; M.R.E.: Visualization, Writing—Review & Editing; M.C.R.: Conceptualization, Supervision, Writing—Review & Editing; I.M.: Conceptualization, Supervision, Writing—Review & Editing.

## Funding

S.J. discloses support for the research of this work from China Scholarship Council (CSC, No. 202106340024). J.A. acknowledges support from an Alexander von Humboldt Research Award. Open Access funding enabled and organized by Projekt DEAL.

## Competing interests

The authors declare no competing interests.
