## [Transparent Peer Review file · Communications Biology]

Rapid evolution in necromass use under resource limitation reduces persistence in producer-decomposer microbial biospheres

Corresponding Author: Professor India Mansour

Version 0:

Reviewer comments:

Reviewer #1

(Remarks to the Author)

Review: Jiang et al, Communications Biology

Overall Summary

In this study, the authors investigate whether evolution in a resource-limited, spatially structured environment influences the persistence of a synthetic microbial biosphere under resource-limited conditions. To address this question, the authors evolve *E. coli* populations for 60 days in closed systems without nutrient supplementation, requiring the populations to sustain themselves via necromass recycling. Half of the monocultures were maintained in homogenized conditions, and the other half contained quartz sand to introduce spatial structure. Following monoculture, the evolved *E. coli* populations were subsequently co-cultured with *C. reinhardtii* in a synthetic microbial biosphere that similarly depends on efficient nutrient cycling to sustain the experimental community.

The authors tracked the abundance of both species and the phenotypic evolution of *E. coli* across multiple co-culture timepoints to elucidate potential mechanisms contributing to the biospheres' stability. Specifically, the authors phenotyped the evolved *E. coli* populations to assess biofilm formation, motility, and carbon source utilization. Ultimately, while the study identifies phenotypic variation among the adapted populations that is influenced by both the initial genetic background and the spatial structure pre-treatment, the authors were unable to establish a clear link between the evolved populations and improved biosphere longevity. Particularly, the systems seeded with WT progenitor *E. coli* demonstrated greater longevity than those containing the evolved strains. There are other concerns about the model, such as the fact that *C. reinhardtii* are facultative heterotrophs. It's also unclear if *C. reinhardtii* can compete for the necromass in the system as well. This further raises questions about how the populations are assessed for abundance in co-culture and if chlorophyll fluorescence loss necessarily means the *C. reinhardtii* population is gone, especially as the authors show "that individual components (e.g., *E. coli* or *C. reinhardtii*) could still persist after system-level failure". Finally, the authors never really explain what ultimately leads to the biosphere's collapse - is it a lack of usable resources? pH? Buildup of toxic metabolic byproducts? - which left me a little unsatisfied.

My more general comments are as follows:

Comments

In general, the introduction should be edited for clarity. Several paragraphs are quite short, and combining them will increase readability. For example, the first paragraph contains only two sentences.

The claim made in line 31 is well established, so the word "probably" should be removed. Instead, I suggest that the authors consider including additional citations discussing long-term starvation studies in *E. coli*. This will help communicate the current state of the literature.

When introducing the two *E. coli* genotypes in line 52, the authors should include more information on how these genotypes differ. Since the results are strain-specific, it's impossible to know why or what to expect without a thorough description of the two genotypes and their differences.

The authors use chlorophyll fluorescence as a proxy for biosphere stability (line 141). However, *C. reinhardtii* is a facultative heterotroph and may be able to use acetate produced by *E. coli* (as a product of carbon metabolism) as a sole source of carbon. See [https://doi.org/10.1016/0032-9592\(94\)80064-2](https://doi.org/10.1016/0032-9592(94)80064-2). *C. reinhardtii*'s facultative heterotrophy should be addressed in

the introduction and should be altered in Figure 1A. Additionally, it should be clearly noted in this Figure that *E. coli* can consume its own necromass and that *C. reinhardtii* may also utilize necromass.

Figure 2A is very convoluted. Sankey diagrams are used to display flow through a system, but here the authors are using one instead of a table or calculating the correlation coefficients between phenotypes. A Sankey diagram may be effective for the first part of the figure, where the authors communicate the breakdown between strains, treatments, and samples. However, for the latter part of the diagram displaying results of the phenotypes, it is hard to follow and is not the best way to visualize these data. I find the sample names are also difficult to follow, and the authors should define these names in the figure legend for easy reference.

In Figure 2B, it is not clear from the figure legend which genotypes are represented in the plot (B, W, or both?). The X-axis should be on a continuous scale, based on the time of harvest, instead of ordinal as it currently stands.

The data visualization in Figure 3A is nice. It is intuitive and clearly communicates the specificity and range of the tested populations. However, it would be nice to see the raw data in a supplementary figure. Given that the data represented in this figure is only the B strain genotype, the authors should clearly note this in the Figure title.

In the text (lines 127-129), it states that HMG-derived populations evolve the greatest increase in resource capacity across the eight tested carbon sources, with 3B referenced. However, when accounting for the variance values included in Figure 3B, the averages all overlap (except LB and CreiNecro). So, this claim is not adequately supported. Moreover, it is never stated in Figure 3B what variance metric is being used (ex., standard error, 95% CI, etc.).

The authors cite Figure 2B in the text before Figure 2A. Likewise, Figure 3B is cited before Figure 2A.

In the methods, the authors state that they used Congo red indicator plates for curli and Congo red agar plates for biofilm. The authors need to elaborate on how these two plates differ and how they distinguish between curli expression and biofilm.

The experimental design shown in Extended Figure 1 is overwhelming. The images are minuscule, and it leaves me more confused about the methods. At a minimum, it needs to be scaled down to convey only the most important of steps. Otherwise, simply describing these processes in the methods section should be sufficient.

Lines 191-195: The authors state that a significant conclusion of the study is a trade-off between motility and biofilm formation with respect to resource acquisition, as well as a negative correlation between enhanced resource utilization capacity and co-culture system persistence. It would be nice if they supported these claims by plotting them. Moreover, biofilm and motility are presented in the figures on a binary scale throughout the manuscript when these are quantitative traits, and should be plotted/analyzed as such.

The discussion is long and exhaustive and should be shortened.

Reviewer #2

(Remarks to the Author)

I co-reviewed this manuscript with one of the reviewers who provided the listed reports. This is part of the Communications Biology initiative to facilitate training in peer review and to provide appropriate recognition for Early Career Researchers who co-review manuscripts.

Reviewer #3

(Remarks to the Author)

This manuscript describes the changes that bacterial cells undergo when incubated in a closed system. Two different strains of *E. coli* were incubated for 60 days in either a structured or unstructured environment, with only saline, meaning that continued growth would rely on released cell detritus. Then a variety of phenotypic measurements were taken, including growth on various carbon sources and growth in another (structured) enclosed system in the presence of a photosynthetic algae. This manuscript tries to address some very interesting questions: how does adapting in different environmental structures affect metabolic genotypes? How do cells sustain themselves under nutrient-limiting conditions? How does genotype affect the adaptations that organisms gain in response to various environments?

Unfortunately, I think this manuscript suffers from trying to answer too many of these questions at once, and therefore it is a bit muddled and not as strong in the conclusions as it could be. For instance, the addition of a second genotype is not particularly helpful here – there are differences seen in phenotypic changes after the initial incubation, but the second strain isn't used for any downstream analysis, so it is unclear how helpful its inclusion is.

It would also strengthen the manuscript if the authors discussed more about the different selections the bacteria may be going through between the initial monocultures and the phenotypic or other growth assessments. Freezing/thawing, outgrowth in different environments/media after the initial incubation, etc., all can contribute to selection of specific genotypes/phenotypes. It would also be helpful to have more detail for some of these instances, for instance the

“preconditioning culture” mentioned in line 76.

Lastly, because the authors do phenotypic testing in static, but homogenous cultures, whereas they do co-culturing in heterogenous environments, but which are different than the original biospheres, it is unclear how well these relate to each other. While there is some statistical correlation, the number of biological replicates is very small, and so strong conclusions are hard to draw. It might be more helpful for the authors to focus more on one of these factors and have higher replication in order to be more confident about patterns.

Minor suggestions:

It would be helpful to see a table or similar of the phenotypes discussed in regards to the original strains.

Line 40: Did the authors mean failure? If so, it would be helpful to define what that means in this context.

Line 42: It would be helpful to define the specifics of “matter exchange” and “spatial interactions” for this context.

Line 71: It would be helpful to refer to figure 1 again here.

Line 88: I would make sure its clear this is a trend, but not a significant difference.

Line 145: Cells instead of components?

Line 371: SP-SDS method needs a citation here. Is the correlation between OD600 and CFU maintained throughout the experiment? Or only during exponential growth?

Line 402: Were the microbes grown together or separately?

Line 414: traits; where were these overnight cultures started from?

Line 463: Was the 1:1 ratio for volume or cells?

Line 480: Definition of GAMs would be helpful

Several typos, i.e.; line 41: closed instead of close; line 47: trends instead of treads

Figures:

For Figure 2, the colors are a bit confusing – at first they indicate genotype, but then later in the figure they are used to indicate environment. It might be helpful to use different colors (this goes for other figures as well).

Figure 2 and Extended Data Table 1 needs abbreviations defined.

Supplementary Figures 4-7: Needs a description of all of the colors

Figure 3A would be more understandable with more explanation.

Figure 4C – what is the y-axis here?

Reviewer #4

(Remarks to the Author)

In this study the authors carry out an evolution experiment with *E. coli* and *Chlamydomonas reinhardtii* in monoculture and co-culture. The monoculture phase propagated *E. coli* in heterogenous or homogenous conditions, and the coculture conditions involved taking the wt, HTG and HMG populations for coculture with *C. reinhardtii*. What makes this study unique is the experiments are carried out in closed microbial biospheres. Since this study is well connected with the microbial experimental evolution literature and theory, the results are discussed in the light of the expectations of the most relevant results of open system biology. Closed system biology is massively understudied, and this important contribution makes a significant advance. The findings highlight that a lot of work needs to be done to understand how ecological communities will evolve in closed systems. The authors are, hopefully, pioneers of a new field of robust microbial biosphere experiment and theory.

The paper is very well written with almost no errors, the figures are beautifully presented. My comments below:

Figure 2. Panel D – should these biofilm measurements be normalised by the number of cells in the culture? I am concerned the increase in raw OD570 could be due to higher cell counts rather than greater per cell biofilm forming capacity.

Line 222 I agree with the authors that slower divergence of *E. coli* W CP002185 from its ancestor suggests that it experienced weaker selection pressures, however the explanation could be related to diminishing returns epistasis. Many evolution experiments have shown that in situations such as described here, where you have two different *e. coli* strains evolving in the same conditions for the same period of time, you would expect the more poorly adapted strain to evolve a greater increase in fitness during that period. The idea is that better adapted strains have less room for improvement. The authors mentioned that *E. coli* W CP002185 has a wider carbon utilisation range, supporting that it was better adapted to the conditions of the experiment than the K12 derived strain. If the authors agree, perhaps they could mention this.

Line 230 The authors refer to “genomic streamlining” incorrectly - Genome streamlining is the evolutionary process where a genome reduces its size by shedding unnecessary genes to increase fitness. Here is a recent example of its use:
<https://doi.org/10.1093/gbe/evae250>

Line 292 I think that one of the most interesting findings is that the *E. coli* that had become better adapted to efficient nutrient cycling were not better able to enhance system persistence. Your possible explanation is not quite clear though – are you saying that since evolution in monoculture drove the loss of motility, this was the reason that the wt *E. coli* was a better “partner” for *C. reinhardtii*? This just needs some clarification in the writing.

Version 1:

Reviewer comments:

Reviewer #1

(Remarks to the Author)

It is appreciated that the authors have made substantial revisions to the manuscript to address concerns regarding the study's model, data visualization, and conclusions. These revisions have significantly improved the quality of the study. However, given the extent of the changes, we still have several remaining concerns, as well as a few additional issues arising from the revised text.

To address the concerns regarding the model, the authors now explicitly stated in the manuscript that *C. reinhardtii* can compete with *E. coli* for necromass and that the loss of fluorescence reflects a collapse of primary production rather than necessarily indicating extinction. This distinction is important as it highlights key caveats that are critical to interpreting the study's conclusions.

The authors have also revised the manuscript to clearly state that the WT *E. coli* strain demonstrates greater longevity than the evolved strains and that the initial hypothesis is not supported by the data. We agree with the authors that the removal of the *E. coli* W strain improves the focus of the key findings of the study.

It is also appreciated that the authors made efforts to improve readability by substantially revising the introduction and condensing the discussion. The addition of literature discussing LTSP and GASP helps better situate the study within the current state of the field. However, the revisions also introduce several issues related to metabolism that should be addressed. These concerns are outlined below:

Line 52-53: "E. coli respire CO₂ from necromass decomposition for photosynthetic assimilation by *C. reinhardtii*, which in turn maintains aerobic conditions via O₂ release." This needs to be reworded because, as written, it initially sounds like the authors are claiming *E. coli* is carbon-fixing (respire CO₂), which they cannot do. It is also unclear if there is evidence that *E. coli* aren't fermenting necromass under these conditions. It would be more accurate to write: *E. coli* metabolizes necromass, releasing CO₂ for photosynthetic.....

Line 211-212 and 220-221: Fermentation should be mentioned as a possibility here as well.

The revisions to the manuscript figures help in improving the clarity and effectiveness of the data visualization. With that being said, there are still remaining concerns.

The use of the Sankey diagram in Figure 2B remains uninformative as it does not allow tracking of populations through the diagram. If the goal is to show phenotypic shifts, the data cannot be combined for each trait. Each track must remain separate (ie. separate positive and negatives for each treatment, WT, HTG, and HMG). The table is far more practical and it is recommended that the Sankey diagram be eliminated from the final version.

Finally, the use of the qualitative method for measuring motility is not ideal, particularly when analyzing evolved phenotypes in a small number of populations. Nevertheless, given that the data are collected, it is recommended that the analysis proceeds but with the limitation acknowledged.

Reviewer #2

(Remarks to the Author)

I co-reviewed this manuscript with one of the reviewers who provided the listed reports. This is part of the Communications Biology initiative to facilitate training in peer review and to provide appropriate recognition for Early Career Researchers who co-review manuscripts.

Reviewer #3

(Remarks to the Author)

The authors have addressed all comments. Thanks to the authors for their thoughtful responses!

Reviewer #4

(Remarks to the Author)

The authors have addressed all of my concerns, and I have no further comments. Congratulations on a really nice study.

Manuscript Title: “Rapid evolution in necromass use under resource limitation reduces persistence in producer-decomposer microbial biospheres”

We thank the reviewers for their insightful comments and suggested revisions, which have significantly improved the quality of our work. Below, we provide a point-by-point response to the reviewers’ comments.

Reviewer #1 (Remarks to the Author):

Review: Jiang et al, Communications Biology

Comment:

Overall Summary

In this study, the authors investigate whether evolution in a resource-limited, spatially structured environment influences the persistence of a synthetic microbial biosphere under resource-limited conditions. To address this question, the authors evolve *E. coli* populations for 60 days in closed systems without nutrient supplementation, requiring the populations to sustain themselves via necromass recycling. Half of the monocultures were maintained in homogenized conditions, and the other half contained quartz sand to introduce spatial structure. Following monoculture, the evolved *E. coli* populations were subsequently co-cultured with *C. reinhardtii* in a synthetic microbial biosphere that similarly depends on efficient nutrient cycling to sustain the experimental community.

The authors tracked the abundance of both species and the phenotypic evolution of *E. coli* across multiple co-culture timepoints to elucidate potential mechanisms contributing to the biospheres’ stability. Specifically, the authors phenotyped the evolved *E. coli* populations to assess biofilm formation, motility, and carbon source utilization. Ultimately, while the study identifies phenotypic variation among the adapted populations that is influenced by both the initial genetic background and the

spatial structure pre-treatment, the authors were unable to establish a clear link between the evolved populations and improved biosphere longevity. Particularly, the systems seeded with WT progenitor *E. coli* demonstrated greater longevity than those containing the evolved strains. There are other concerns about the model, such as the fact that *C. reinhardtii* are facultative heterotrophs. It's also unclear if *C. reinhardtii* can compete for the necromass in the system as well. This further raises questions about how the populations are assessed for abundance in co-culture and if chlorophyll fluorescence loss necessarily means the *C. reinhardtii* population is gone, especially as the authors show “that individual components (e.g., *E. coli* or *C. reinhardtii*) could still persist after system-level failure”. Finally, the authors never really explain what ultimately leads to the biosphere's collapse - is it a lack of usable resources? pH? Buildup of toxic metabolic byproducts? - which left me a little unsatisfied.

Response:

Thank you for this accurate summary and for raising these critical points. We agree that the absence of a simple, positive relationship between decomposer adaptation and biosphere persistence is a core and counterintuitive outcome of our study. We have substantially revised the Discussion and Introduction to explicitly address these concerns.

First, we now emphasize that our data do not support the initial hypothesis that enhanced necromass utilization in *E. coli* necessarily prolongs system persistence. Instead, we explicitly frame this as an evolutionary trade-off: phenotypic adaptations beneficial in monocultures under extreme resource limitation can become maladaptive in a producer-decomposer coexistence context (see page 9 lines 196-198; page 13, lines 282-284).

“We hypothesized that efficient necromass recycling by the decomposer would prolong

system persistence by expanding the cycling nutrient pool. However, phenotypic shifts acquired during resource-limited monoculture unexpectedly accelerated collapse in subsequent co-cultures (Figure 6B)."

"In conclusion, resource-limited adaptation can yield counterintuitive outcomes: traits enhancing decomposer performance in isolation may diminish producer-decomposer system persistence during species interactions."

Second, in the revised Discussion, we explicitly acknowledge that *C. reinhardtii* is a facultative heterotroph that competes with *E. coli* for organic substrates, including necromass-derived compounds. We propose that pre-adapted *E. coli* populations with expanded carbon utilization capabilities may intensify this competition, thereby suppressing algal activity and accelerating system-level collapse (see page 10 lines 199-207).

*"This outcome may be explained by several non-mutually exclusive mechanisms: First, intensified resource competition likely played a key role. While *E. coli* and *C. reinhardtii* engage in reciprocal material exchange (CO_2 /inorganic nutrients for organic carbon)^{33,34}, *C. reinhardtii* is a facultative heterotroph that competes for organic substrates²⁵. Pre-adapted *E. coli* (particularly HMG-derived populations) exhibited enhanced carbon utilization, potentially outcompeting *C. reinhardtii* for necromass-derived compounds. Although starved *E. coli* incompletely metabolizes endogenous compounds and necromass³⁵, potentially releasing metabolites such as aromatic amino acids (e.g., phenylalanine) that could be utilized by *C. reinhardtii*³⁶, its aggressive resource acquisition likely overwhelms this potential benefit and may suppress *C. reinhardtii*."*

Third, we have clarified the interpretation of chlorophyll fluorescence. We now explicitly state that the loss of fluorescence reflects a collapse of primary production

(system failure), not necessarily the immediate extinction of *C. reinhardtii*. A loss in ongoing carbon cycling (which could but doesn't need to align with a loss of one or more system populations) is our indicator for a system which can no longer self-sustain. This aligns with our destructive sampling data showing individual species persisting after the system ceases to function photosynthetically (see page 2 lines 42-44; page 3 lines 54-56; page 6 lines 132-133).

“These experimental systems allow laboratory-scale observation of system failure, which we defined as the collapse of primary production, reflected by the loss of photosynthetic activity²².”

“System persistence is monitored at the ecosystem level via photosynthetic activity. Accordingly, we consider the system to have failed once this activity ceases, regardless of temporary species persistence on alternative substrates.”

“All co-cultures ceased photosynthetic activity within 10 days, indicating system-level failure (Supplementary Figure 6).”

Finally, we have rewritten the discussion to include several non-mutually exclusive mechanisms driving collapse, including intensified resource competition, disrupted spatial coupling due to loss of bacterial motility, accumulation of inhibitory metabolites, and oxygen limitation following diminished algal photosynthesis (see page 10 line 199- page 11 line 232). We clarify that collapse likely stems from a synergy of these factors rather than a single resource depletion.

*“This outcome may be explained by several non-mutually exclusive mechanisms: First, intensified resource competition likely played a key role. While *E. coli* and *C. reinhardtii* engage in reciprocal material exchange (CO_2 /inorganic nutrients for organic carbon)^{33,34}, *C. reinhardtii* is a facultative heterotroph that competes for organic*

substrates²⁵. Pre-adapted *E. coli* (particularly HMG-derived populations) exhibited enhanced carbon utilization, potentially outcompeting *C. reinhardtii* for necromass-derived compounds. Although starved *E. coli* incompletely metabolizes endogenous compounds and necromass³⁵, potentially releasing metabolites such as aromatic amino acids (e.g., phenylalanine) that could be utilized by *C. reinhardtii*³⁶, its aggressive resource acquisition likely overwhelms this potential benefit and may suppress *C. reinhardtii*.

Second, the loss of motility in pre-adapted *E. coli* likely disrupts spatial coupling with the producers. Bacterial motility facilitates proximity to *C. reinhardtii*, enabling efficient aerobic respiration and nutrient exchange^{37,38}. While motile populations (WT progenitor and HTG-derived populations HTG1/HTG2) could forage on *C. reinhardtii* exudates, HMG-derived populations lost motility and increased biofilm formation. This sessile lifestyle likely sequesters nutrients within the biofilm matrix. Since diffusion is a slow process, this sequestration privatizes and hoards shared nutrient resources (the “commons”) for immediate local benefit, while weakening interaction strength³⁹. This preferential resource retention limits reciprocal nutrient exchange with *C. reinhardtii*. Since algal proliferation depends on the accumulation of CO₂ and recycled nutrients⁴⁰, this limitation reduces algal photosynthetic activity and potentially depletes oxygen availability. Consequently, due to this lack of oxygen, *E. coli* may be forced into less efficient anaerobic metabolism. These combined effects create a “tragedy of the commons” (wherein individual optimization undermines collective stability) and likely impede coexistence.

Third, system collapse may involve inhibitory metabolite accumulation. The rapid decline in photosynthetic activity suggests *C. reinhardtii* suppression, which could also be driven by incomplete necromass degradation or the buildup of toxic metabolic products⁴¹. Such metabolic by-products could directly inhibit algal physiology, contributing to system failure.

Consequently, collapse likely stems from a synergy of resource competition, disrupted coupling, and metabolic inhibition rather than a single resource depletion.

Finally, evolutionary mismatches between monoculture adaptation and co-culture conditions may have accelerated collapse. In small, fragmented populations, genetic drift can fix deleterious mutations or cause the loss of beneficial alleles⁴². Traits that are beneficial in monoculture can become detrimental in co-culture, as observed in shifting selection pressures in stationary-phase batch cultures⁴³. The synergy of drift-driven maladaptation and rapid resource exhaustion likely contributed to population extinction⁴⁴.”

Comment:

My more general comments are as follows:

Comments

In general, the introduction should be edited for clarity. Several paragraphs are quite short, and combining them will increase readability. For example, the first paragraph contains only two sentences.

Response:

Thank you for your valuable feedback. We agree that the previous introduction suffered from overly short paragraphs. The section has been substantially revised to improve flow. We merged the first two paragraphs to connect the general context of nutrient scarcity with necromass recycling. Similarly, the third and fourth paragraphs were combined to integrate the description of the experimental system (microbial biospheres) with the introduction of the model organisms (see page 2 line 27-page 3 line 56).

“Microorganisms frequently encounter nutrient scarcity across diverse environments¹⁻

⁵. To survive, they employ strategies such as altering morphology^{6,7}, modulating motility⁸, enhancing adhesion⁹, simplifying metabolic pathways¹⁰, or entering dormancy¹¹ and growth arrest¹². Notably, many bacteria persist for decades in energy-limited environments by recycling necromass, the organic remnants of dead cells¹³. In *Escherichia coli* (*E. coli*), long-term stationary phase (LTSP) populations endure extreme nutrient depletion through density-dependent death and cellular debris recycling¹⁴. This process involves continuous adaptation characterized by genetic parallelism¹⁵, mutations in global regulators (e.g., *rpoS*)¹⁶, and a shift toward slow growth and stress resistance¹⁷, often resulting in a growth advantage in stationary phase (GASP) phenotype¹⁸. Despite these known survival mechanisms, the ecological consequences of starvation-induced adaptation, particularly its impact on interspecies interactions and ecosystem stability are underexplored and unclear. Furthermore, the interplay between spatial structure (critical for promoting genetic diversity¹⁹, species coexistence²⁰, and ecosystem sustainability²¹) and microbial adaptation under prolonged resource scarcity requires further exploration.

To investigate microbial ecosystem dynamics under controlled, resource-limited conditions, we employed microbial biospheres^{21,22}. These experimental systems allow laboratory-scale observation of system failure, which we defined as the collapse of primary production, reflected by the loss of photosynthetic activity²². Specifically, these biospheres are matter-closed but energy-open (light), necessitating internal carbon and nutrient cycling. This matter-closure minimizes external interference, such as species immigration, and requires microbial regulation of the atmosphere for persistence. Unlike open systems like chemostats²³, microbial biospheres lack nutrient inputs, requiring the evolution of self-contained nutrient cycles. Standardized replication of the biospheres allows for precise monitoring to distinguish deterministic trends from stochastic fluctuations²³. To mimic the core carbon cycle, we co-cultured the autotroph *Chlamydomonas reinhardtii* (*C. reinhardtii*)²⁴ and decomposer *E. coli* MC4100. *E. coli* respire CO_2 from necromass decomposition for photosynthetic assimilation by *C. reinhardtii*, which in turn maintains aerobic conditions via O_2 release. Interactions may

also include mixotrophy, as C. reinhardtii may utilize E. coli metabolites such as acetate²⁵. System persistence is monitored at the ecosystem level via photosynthetic activity. Accordingly, we consider the system to have failed once this activity ceases, regardless of temporary species persistence on alternative substrates.”

Comment:

The claim made in line 31 is well established, so the word “probably” should be removed. Instead, I suggest that the authors consider including additional citations discussing long-term starvation studies in *E. coli*. This will help communicate the current state of the literature.

Response:

We have removed the word “probably” as advised. Additionally, we have expanded this section by citing recent studies on *E. coli* long-term stationary phase (LTSP) and the GASP phenotype, highlighting the well-established nature of bacterial survival via necromass recycling (see page 2 lines 29-35).

“Notably, many bacteria persist for decades in energy-limited environments by recycling necromass, the organic remnants of dead cells¹³. In Escherichia coli (E. coli), long-term stationary phase (LTSP) populations endure extreme nutrient depletion through density-dependent death and cellular debris recycling¹⁴. This process involves continuous adaptation characterized by genetic parallelism¹⁵, mutations in global regulators (e.g., rpoS)¹⁶, and a shift toward slow growth and stress resistance¹⁷, often resulting in a growth advantage in stationary phase (GASP) phenotype¹⁸.”

Comment:

When introducing the two *E. coli* genotypes in line 52, the authors should include more information on how these genotypes differ. Since the results are strain-specific, it's

impossible to know why or what to expect without a thorough description of the two genotypes and their differences.

Response:

To streamline the narrative and address comments regarding the complexity of the data presentation, we have removed all content related to the second strain (*E. coli* W) from the manuscript. This strain exhibited a weaker evolutionary response, and its inclusion did not strengthen the core conclusions. The revised manuscript now focuses solely on *E. coli* MC4100. This allows us to focus the paper on how environmental conditions (spatial structure) shape phenotypic evolution without the added variable of strain-specific genetics.

Comment:

The authors use chlorophyll fluorescence as a proxy for biosphere stability (line 141). However, *C. reinhardtii* is a facultative heterotroph and may be able to use acetate produced by *E. coli* (as a product of carbon metabolism) as a sole source of carbon. See [https://doi.org/10.1016/0032-9592\(94\)80064-2](https://doi.org/10.1016/0032-9592(94)80064-2). *C. reinhardtii*'s facultative heterotrophy should be addressed in the introduction and should be altered in Figure 1A. Additionally, it should be clearly noted in this Figure that *E. coli* can consume its own necromass and that *C. reinhardtii* may also utilize necromass.

Response:

Thank you for this point. We have added a description of *C. reinhardtii* as a facultative heterotroph to the Introduction and Discussion (see page 3 lines 53-54 and page 10 lines 200-202). The text now clearly describes the metabolic interactions, including potential mixotrophy and competition for necromass.

“Interactions may also include mixotrophy, as C. reinhardtii may utilize E. coli metabolites such as acetate²⁵.”

“While E. coli and C. reinhardtii engage in reciprocal material exchange (CO₂/inorganic nutrients for organic carbon)^{34,35}, C. reinhardtii is a facultative heterotroph that competes for organic substrates²⁵.”

Furthermore, Figure 1 (page 23) has been completely redesigned. To avoid the misleading overemphasis on specific carbon flows in the original schematic, the new figure focuses on the experimental workflow and the specific phenotypes tested before and after evolution.

Comment:

Figure 2A is very convoluted. Sankey diagrams are used to display flow through a system, but here the authors are using one instead of a table or calculating the correlation coefficients between phenotypes. A Sankey diagram may be effective for the first part of the figure, where the authors communicate the breakdown between strains, treatments, and samples. However, for the latter part of the diagram displaying results of the phenotypes, it is hard to follow and is not the best way to visualize these data. I find the sample names are also difficult to follow, and the authors should define these names in the figure legend for easy reference.

Response:

We have simplified the Sankey diagram (now Figure 2B, page 26) significantly by removing the second strain (W). We have also added a comprehensive table (Table 1, page 24) detailing the phenotypic changes for clear comparison. We retained the simplified Sankey diagram because it intuitively visualizes the divergence between the homogeneous (HMG) and heterogeneous (HTG) treatments compared to the wild-type

progenitor, specifically highlighting that HMG-derived populations undergo more drastic phenotypic shifts. Sample names have been standardized (e.g., HTG1, HMG1) and are explicitly defined in the figure legend.

Comment:

In Figure 2B, it is not clear from the figure legend which genotypes are represented in the plot (B, W, or both?). The X-axis should be on a continuous scale, based on the time of harvest, instead of ordinal as it currently stands.

Response:

This figure (now Figure 2A, page 26) now represents data solely from *E. coli* MC4100. Regarding the x-axis: because our monoculture system is a closed system (sealed for 60 days), continuous sampling was not possible without compromising the system's integrity. Therefore, data points exist only for the "inoculum" and the "harvest" (day 60). We have clarified this information in the figure caption to explain why an ordinal scale is used (see page 26 lines 497-499).

"Log₁₀-transformed viable cell counts of Escherichia coli (E. coli) at the initiation and upon destructive harvest after 60 days of monoculture in two distinct systems (HTG: spatially heterogeneous; HMG: spatially homogeneous)."

Comment:

The data visualization in Figure 3A is nice. It is intuitive and clearly communicates the specificity and range of the tested populations. However, it would be nice to see the raw data in a supplementary figure. Given that the data represented in this figure is only the B strain genotype, the authors should clearly note this in the Figure title.

Response:

We have replotted the full growth parameters for *E. coli* MC4100 across all carbon sources in Supplementary Figures 2, 3, and 4. The caption for Figure 3A (page 28) has been updated to explicitly state that these data represent the area under the curve (AUC) for *E. coli* MC4100 (see page 28 lines 516-523).

“Comparison of the area under the curve (AUC) for growth curves on eight carbon sources between the wild-type progenitor (WT) and three evolved populations from each monoculture system (HTG: spatially heterogeneous; HMG: spatially homogeneous). Data were normalized to the AUC of the WT and log₁₀-transformed. Consequently, the AUC for the WT is 0 for each carbon source; values > 0 indicate an increase relative to WT, while values < 0 indicate a decrease. CreiNecro and EcoNecro denote the necromass of Chlamydomonas reinhardtii and Escherichia coli, respectively. The original AUC data are presented as beeswarm plots in Supplementary Figures 2-4.”

Comment:

In the text (lines 127-129), it states that HMG-derived populations evolve the greatest increase in resource capacity across the eight tested carbon sources, with 3B referenced. However, when accounting for the variance values included in Figure 3B, the averages all overlap (except LB and CreiNecro). So, this claim is not adequately supported. Moreover, it is never stated in Figure 3B what variance metric is being used (ex., standard error, 95% CI, etc.).

Response:

According to the Dunn’s test results, HMG-derived populations showed a significantly higher area under the curve (AUC, our metric for utilization capacity) than the wild-type progenitor on all carbon sources except xylose (see Supplementary Figures 2-4 for

detailed statistical breakdown). The text has been revised to reflect the statistical significance more precisely (see page 6 lines 119-124).

“HMG-derived populations exhibited a significantly higher resource utilization capacity than the WT progenitor on seven of the eight tested carbon sources (all except xylose) (Figure 3B; statistical details are provided in Supplementary Figures 2-4). While HTG-derived populations also showed increased capacity relative to WT progenitor on most carbon sources (excluding xylose), the magnitude of increase was consistently less pronounced than that of the HMG-derived populations.”

We have also updated the Figure 3B caption to explicitly state that values represent mean \pm standard error (see page 28 lines 525-526).

“Values are displayed as mean \pm standard error (SE).”

Comment:

The authors cite Figure 2B in the text before Figure 2A. Likewise, Figure 3B is cited before Figure 2A.

Response:

We have reorganized the figures and the text to ensure they are cited in strict numerical and alphabetical order.

Comment:

In the methods, the authors state that they used Congo red indicator plates for curli and Congo red agar plates for biofilm. The authors need to elaborate on how these two plates differ and how they distinguish between curli expression and biofilm.

Response:

We agree that distinguishing these assays is crucial. We have updated the Methods section to clarify: Congo Red Agar (CRA) uses a high-sucrose, nutrient-rich formulation to induce general extracellular polymeric substance (EPS) production, visualizing global biofilm matrix formation (black colonies). (see page 19 lines 400-405).

“The high concentration of sucrose in the medium was used to promote the production of extracellular polymeric substances. Plates were incubated at 37°C, and results were recorded at 24 and 48 hours. CRA visualizes extracellular polymeric substances characteristic of biofilms; biofilm formers typically produce black, dry, crystalline colonies, whereas non-formers produce red or pink colonies²⁸ (Supplementary Figure IIC).”

Conversely, Congo Red Indicator (CRI) uses a low-sugar formulation to minimize non-specific polysaccharide interference, specifically detecting curli fimbriae (amyloid) expression via a brown, dry, and rough (BDAR) phenotype. (see page 18 line 393-page 19 line 397).

“CRI plates utilize a low sugar formulation, which minimizes interference from excessive, nonspecific polysaccharide production and specifically focuses on curli fimbriae expression. Colonies expressing curli fimbriae bind both the Congo Red and Brilliant Blue R dyes and exhibit a brown, dry, and rough (BDAR) morphology, while non-expressing colonies remain smooth and white (SAW)²⁷ (Supplementary Figure IIB).”

Comment:

The experimental design shown in Extended Figure 1 is overwhelming. The images are

minuscule, and it leaves me more confused about the methods. At a minimum, it needs to be scaled down to convey only the most important of steps. Otherwise, simply describing these processes in the methods section should be sufficient.

Response:

We have removed this figure (formerly Extended Data Fig. 1) from the revised manuscript to reduce clutter and confusion.

Comment:

Lines 191-195: The authors state that a significant conclusion of the study is a trade-off between motility and biofilm formation with respect to resource acquisition, as well as a negative correlation between enhanced resource utilization capacity and co-culture system persistence. It would be nice if they supported these claims by plotting them. Moreover, biofilm and motility are presented in the figures on a binary scale throughout the manuscript when these are quantitative traits, and should be plotted/analyzed as such.

Response:

Thank you for this constructive suggestion. We agree that visualizing these correlations and providing more quantitative depth would strengthen the paper.

Motility: We acknowledge that motility is a quantitative trait. However, due to experimental constraints, the motility assay we employed (stab inoculation into semi-solid media) is an established qualitative method with a binary outcome (motile / non-motile). This method was appropriate for our scientific question as it effectively distinguished evolved populations from the wild-type progenitor. Although motility is physiologically a quantitative trait, our experimental design aimed to capture a

phenotypic shift of ecological relevance.

Biofilm: We employed a quantitative approach to assess biofilm formation capacity: the tissue culture plate (TCP) assay with crystal violet staining, which yields continuous absorbance readings (OD570). Specifically, Figures 2C (page 26) and Supplementary Figure 1 explicitly present the continuous OD570 data quantified over time via the TCP assay, allowing direct comparison of biofilm formation capacity across treatment groups.

Correlation: We replaced the original Figure 5B with a new correlation plot (Figure 5C, page 32) and a full correlation heatmap (Supplementary Figure 10). These plots use the quantitative TCP biofilm data and quantitative growth parameters (AUC) to demonstrate the negative correlation between resource utilization/biofilm and system persistence (time to chlorophyll fluorescence loss).

Survival Analysis: For the Kaplan-Meier survival analysis (Figure 5A), we retained categorical bins (e.g., none/weak/moderate/strong) because this statistical method requires grouped data. In Figure 4E, we also utilized categorical bins to more clearly illustrate the direction and magnitude of changes in biofilm formation capacity across populations. The criteria for these categories are defined in the Methods (see page 19 lines 413-417).

“Biofilm-forming ability was categorized based on OD570 values (ODS) compared to the negative control (ODC): $ODS \leq ODC$ = no biofilm producer, $ODC < ODS \leq (2 \times ODC)$ = low biofilm producer, $2 \times ODC < ODS \leq 4 \times ODC$ = moderate biofilm producer, and $4 \times ODC < ODS$ = strong biofilm producer²⁶. TCP data at 72 hours were used as the primary indicator for biofilm production.”

Comment:

The discussion is long and exhaustive and should be shortened.

Response:

We have restructured and condensed the Discussion into three focused subsections, reducing the word count from 1555 to 1119: (1) Evolutionary trade-offs, (2) Environmental drivers of phenotypic evolution, and (3) Limitations and future perspectives. We removed repetitive text and focused on the key mechanisms (e.g., loss of motility, intensified competition, disruption of spatial coupling) that drive the observed system collapse (see pages 9-14).

Reviewer #2 (Remarks to the Author):

Comment:

I co-reviewed this manuscript with one of the reviewers who provided the listed reports. This is part of the Communications Biology initiative to facilitate training in peer review and to provide appropriate recognition for Early Career Researchers who co-review manuscripts.

Response:

Thank you and your colleague for the many constructive suggestions you have provided for this paper.

Reviewer #3 (Remarks to the Author):

Comment:

This manuscript describes the changes that bacterial cells undergo when incubated in a closed system. Two different strains of *E. coli* were incubated for 60 days in either a structured or unstructured environment, with only saline, meaning that continued growth would rely on released cell detritus. Then a variety of phenotypic measurements were taken, including growth on various carbon sources and growth in another (structured) enclosed system in the presence of a photosynthetic algae. This manuscript tries to address some very interesting questions: how does adapting in different environmental structures affect metabolic genotypes? How do cells sustain themselves under nutrient-limiting conditions? How does genotype affect the adaptations that organisms gain in response to various environments?

Unfortunately, I think this manuscript suffers from trying to answer too many of these questions at once, and therefore it is a bit muddled and not as strong in the conclusions as it could be. For instance, the addition of a second genotype is not particularly helpful here – there are differences seen in phenotypic changes after the initial incubation, but the second strain isn't used for any downstream analysis, so it is unclear how helpful its inclusion is.

Response:

We sincerely thank you for this constructive feedback. We agree that the inclusion of the second strain (*E. coli* W) introduced unnecessary complexity without adding significant depth to the core ecological narrative, particularly given its lack of strong evolutionary response in our experiments. In the revised manuscript, we have removed all data and discussion regarding *E. coli* W. The study now focuses exclusively on *E. coli* MC4100. This streamlines the narrative, preventing potential confusion arising from strain-specific variances, and allows us to focus the manuscript on the central question: how phenotypic adaptation to spatial structure under resource limitation affects subsequent producer-decomposer coexistence.

Comment:

It would also strengthen the manuscript if the authors discussed more about the different selections the bacteria may be going through between the initial monocultures and the phenotypic or other growth assessments. Freezing/thawing, outgrowth in different environments/media after the initial incubation, etc., all can contribute to selection of specific genotypes/phenotypes. It would also be helpful to have more detail for some of these instances, for instance the “preconditioning culture” mentioned in line 76.

Response:

Thank you for raising this important point regarding potential selection pressure during post-monoculture handling. We agree that steps such as freeze-thaw cycles and growth in nutrient-rich LB could, in principle, select for phenotypes different from those favored in the initial resource-limited closed system.

To address this, we have clarified the preconditioning protocol. We expanded the description of the preconditioning culture in the Results (see page 4 lines 72-77) and Methods (see page 15 lines 315-321) for full transparency. We explicitly state that standardized overnight cultures were used to ensure comparable physiological states (approx. 20 generations in LB) prior to testing.

“To ensure reproducible and comparable phenotypic assessments, all recovered E. coli populations were subjected to a standardized preconditioning culture procedure prior to analysis. Briefly, this involved inoculation from glycerol stocks into Luria-Bertani (LB) medium for overnight growth, followed by a second overnight passage, ensuring approximately 20 generations of growth in a consistent physiological state before testing.”

“For subsequent phenotypic and growth assays, frozen glycerol stocks were recovered and used to inoculate a preconditioning culture: a single overnight culture in LB medium. This step ensured consistent and active starting cultures and acclimated cells to standard laboratory conditions after long-term incubation in saline. Following this, a fresh overnight culture was initiated from the preconditioning culture to generate the cells used directly in testing. Thus, between recovery from the closed monoculture system and final assessments, cells underwent approximately 20 generations of growth in rich, homogeneous LB medium.”

We have also added a paragraph to the Discussion acknowledging that the recovery phase introduced different selective pressures (see page 12 lines 262-264).

“First, the approximately 20-generation incubation in LB medium required for diagnostic phenotyping introduced different selective pressures compared to the original closed-system environments (HTG and HMG).”

However, we argue that the observed phenotypes are likely stable genetic adaptations rather than transient physiological states or artifacts of handling. Crucially, while all populations underwent identical handling (freezing and LB preconditioning), only the spatially distinct groups (HTG vs. HMG) exhibited divergent, habitat-specific phenotypes (e.g., loss of motility in HMG). The persistence of these distinct traits despite ~20 generations of growth in rich media suggests they are robust evolutionary changes driven by the 60-day monoculture, not the recovery process (see page 12 lines 255-260).

“The maintenance of distinct phenotypic profiles through approximately 20 generations of recovery and cultivation in nutrient-rich media further supports a genetic basis, as this process would dilute transient epigenetic effects^{54,55}. The lack of reversion to wild-type phenotypes and the continued directional changes during co-culture confirm that

these observed traits represent stable evolutionary responses to their respective environmental conditions.”

Comment:

Lastly, because the authors do phenotypic testing in static, but homogenous cultures, whereas they do co-culturing in heterogenous environments, but which are different than the original biospheres, it is unclear how well these relate to each other. While there is some statistical correlation, the number of biological replicates is very small, and so strong conclusions are hard to draw. It might be more helpful for the authors to focus more on one of these factors and have higher replication in order to be more confident about patterns.

Response:

Thank you for this important point. Regarding the testing environment, we have clarified in the Discussion that the phenotypic assays (conducted in rich, homogeneous media) were intended as standardized “diagnostic tools” to quantify heritable phenotypic changes, rather than to mimic the co-culture environment (see page 12 line 262-page 13 line 266). Our goal was to characterize the fundamental traits (motility, biofilm, carbon use) the bacteria acquired during evolution before asking how those traits played out in the complex, heterogeneous co-culture system.

“First, the approximately 20-generation incubation in LB medium required for diagnostic phenotyping introduced different selective pressures compared to the original closed-system environments (HTG and HMG). We emphasize that this standardized, nutrient-rich assay was designed as a controlled diagnostic tool to quantify heritable fitness changes acquired during long-term selection, not to mimic the co-culture environment.”

Regarding replication, we acknowledge that starting with $n = 3$ independent evolutionary lineages per treatment is a limitation. However, we emphasize two points that bolster our confidence in the results. First, the co-culture phase involved substantial replication ($n = 21$ biospheres per sample, totaling 168 biospheres) to robustly assess the ecological impact of the evolved populations (see page 21 lines 440-443).

*“Each plate included three technical replicates for each *E. coli* + *C. reinhardtii* sample, randomly distributed across the plate. A total of seven plates were prepared, resulting in 21 experimental biosphere replicates per sample and 21 negative control replicates across the entire experiment.”*

Second, despite the small number of lineages, we observed strong phenotypic convergence. For example, all HMG-derived populations lost motility and increased biofilm formation. This high degree of parallelism suggests a strong, deterministic selective pressure rather than stochastic variation (see page 13 lines 266-269). Furthermore, the multi-stage nature of this study—long-term evolution, followed by extensive phenotyping and co-culture—made larger biological replication impractical, yet the observed effects are clear and meaningful within the experimental constraints.

“Second, while limited replication ($n = 3$ per treatment) restricts inferences on among-lineage variability, the robust convergence of traits, such as motility loss in HMG-derived populations, underscores the strength of the observed selection.”

We have updated the Discussion to transparently address these limitations while highlighting the consistency of the evolutionary signal we observed (see above). We also note that future studies with in-situ phenotyping would be valuable (see page 13 lines 274-275).

“(ii) conduct in situ assays to verify phenotypes under authentic biosphere conditions”

Comment:

Minor suggestions:

It would be helpful to see a table or similar of the phenotypes discussed in regards to the original strains.

Response:

Thank you for this suggestion. We have moved the detailed phenotype table from the Extended Data into the main text as Table 1. This table now directly complements the Sankey diagram (see page 26 Figure 2B) to provide a clear overview of the phenotypic changes (see page 24 Table 1).

Comment:

Line 40: Did the authors mean failure? If so, it would be helpful to define what that means in this context.

Response:

We agree that “failure” requires a precise definition in this context. We have revised the Introduction to explicitly define system failure as “the collapse of primary production, reflected by the loss of photosynthetic activity”. (see page 2 lines 42-44; page 3 lines 54-56).

“These experimental systems allow laboratory-scale observation of system failure, which we defined as the collapse of primary production, reflected by the loss of photosynthetic activity²².”

“System persistence is monitored at the ecosystem level via photosynthetic activity.

Accordingly, we consider the system to have failed once this activity ceases, regardless of temporary species persistence on alternative substrates.”

Comment:

Line 42: It would be helpful to define the specifics of “matter exchange” and “spatial interactions” for this context.

Response:

Thank you for this request for precision. We have revised the Introduction to clearly define these terms within our experimental context. Specifically, we clarify that “matter-closed” refers to systems lacking external nutrient inputs or waste removal, necessitating internal cycling and the autonomous maintenance of a livable atmosphere by the enclosed organisms. Furthermore, “spatial interactions” refers to external interference (e.g., immigration), which our closed biospheres exclude to isolate internal dynamics (see page 2 line 44-page 3 line 50).

“Specifically, these biospheres are matter-closed but energy-open (light), necessitating internal carbon and nutrient cycling. This matter-closure minimizes external interference, such as species immigration, and requires microbial regulation of the atmosphere for persistence. Unlike open systems like chemostats²³, microbial biospheres lack nutrient inputs, requiring the evolution of self-contained nutrient cycles. Standardized replication of the biospheres allow for precise monitoring to distinguish deterministic trends from stochastic fluctuations²³.”

Comment:

Line 71: It would be helpful to refer to figure 1 again here.

Response:

We have added a reference to Figure 1 in the text as suggested (see page 4 lines 69-70).

“After 60 days of monoculture in a closed system, we recovered E. coli populations from both HTG and HMG spatial contexts (Figure 1).”

Comment:

Line 88: I would make sure its clear this is a trend, but not a significant difference.

Response:

Thank you for this correction. We have revised the text describing biofilm formation to accurately reflect the statistical results. We now clarify that while there was a clear temporal trend of increasing biofilm formation, the difference between HMG and HTG-derived populations at certain time points was a non-significant trend, though HMG-derived populations were significantly higher than the WT progenitor at 48 hours (see page 4 line 86-page 5 line 91).

“Quantitative crystal violet assays²⁶ for biofilm formation revealed a clear temporal trend, with biofilm abundance increasing over time, where the HMG-derived populations consistently formed the most biofilm and WT progenitor the least across all three incubation periods (Supplementary Figure 1). After 48 hours, the biofilm formation ability of each population peaked, with HMG-derived populations significantly exceeding the WT progenitor but not differing significantly from HTG-derived populations (Figure 2C).”

Comment:

Line 145: Cells instead of components?

Response:

We agree that “components” is ambiguous. We have replaced it with “species” to accurately refer to the *E. coli* and *C. reinhardtii* populations surviving in the biospheres (see page 7 lines 138-139).

“Destructive sampling at days 10, 16, 23, and 28 showed that individual species (e.g., E. coli or C. reinhardtii) could still persist after system-level failure.”

Comment:

Line 371: SP-SDS method needs a citation here. Is the correlation between OD600 and CFU maintained throughout the experiment? Or only during exponential growth?

Response:

We have added the citation for the SP-SDS method (see page 16 lines 335-336).

“OD600 was measured in 96-well plates, and CFU counts were obtained using the SP-SDS method⁵⁶.”

Regarding the OD600/CFU correlation: To ensure the validity of our standard curves, we extended the *E. coli* incubation for standard curve generation to 20 hours to achieve high densities. The resulting standard curves showed excellent linearity ($R^2 > 0.98$) across the range of densities observed in our experiments (see Supplementary Figure 12 and 13). By using 18-hour cultures for all downstream analyses, we ensured the measurements fell within the validated linear range of our standard curve throughout the experiment.

Comment:

Line 402: Were the microbes grown together or separately?

Response:

They were grown separately. We have clarified in the Methods that *C. reinhardtii* and *E. coli* cultures were grown separately before being harvested to create the necromass (see page 17 lines 369-371).

“To evaluate necromass utilization, C. reinhardtii and E. coli cultures were grown separately to the stationary phase, harvested, and autoclaved to create necromass solutions (0.05 mol C L⁻¹).”

Comment:

Line 414: traits; where were these overnight cultures started from?

Response:

We have corrected the typo “straits” to “traits”. We also clarified that overnight cultures used for phenotyping were not inoculated directly from frozen stocks, but were generated via the preconditioning protocol (glycerol stock – overnight culture – fresh overnight test culture) to ensure metabolic consistency (see page 18 lines 383-385).

“To assess E. coli traits at the population level, we used standardized overnight cultures (generated via the preconditioning protocol described in 5.1), which represent a mixture of cells from each evolved population.”

Comment:

Line 463: Was the 1:1 ratio for volume or cells?

Response:

The 1:1 ratio refers to volume. We have clarified this in the text as “1:1 v/v” and explicitly stated that this results in an initial cell count ratio of approximately 1:50 (*C. reinhardtii* to *E. coli*). (see page 20 lines 437-438).

“Equal volumes of these suspensions were mixed (1:1 v/v), yielding a co-culture inoculum with an initial C. reinhardtii to E. coli cell count ratio of 1:50.”

Comment:

Line 480: Definition of GAMs would be helpful

Response:

We have defined GAMs as Generalized Additive Models in the text (see page 21 line 459).

“Generalized additive models (GAMs) were used to process ...”

Comment:

Several typos, i.e.; line 41: closed instead of close; line 47: trends instead of treads

Response:

Thank you for spotting these. We have corrected “close” to “closed” and ensured “trends” is spelled correctly throughout the manuscript.

Comment:

Figures:

For Figure 2, the colors are a bit confusing – at first they indicate genotype, but then

later in the figure they are used to indicate environment. It might be helpful to use different colors (this goes for other figures as well).

Response:

We have completely redesigned the color scheme across all figures to ensure consistency. Gray: Wild-type progenitor (WT); Orange hues: HTG-derived populations; Blue hues: HMG-derived populations. This scheme is now applied consistently in all figures. Additionally, Figure 2 has been simplified by removing the second strain to reduce visual clutter (see page 26 Figure 2).

Comment:

Figure 2 and Extended Data Table 1 needs abbreviations defined.

Response:

We have updated the captions for Figure 2 and the footnotes of Table 1 (formerly Extended Data Table 1) to explicitly define all abbreviations (e.g., CRA, TCP, HTG, HMG) (see page 26 Figure 2 and page 24 Table 1).

Comment:

Supplementary Figures 4-7: Needs a description of all of the colors

Response:

We have redrawn these figures (now Supplementary Figures 2-4) using the standardized color scheme (Gray = WT, Orange = HTG, Blue = HMG) and provided detailed legends describing the color coding and the meaning of the shaded comparison areas.

Comment:

Figure 3A would be more understandable with more explanation.

Response:

We have expanded the caption for Figure 3A to explain the data transformation process (see page 28 lines 516-523).

“(A) Comparison of the area under the curve (AUC) for growth curves on eight carbon sources between the wild-type progenitor (WT) and three evolved populations from each monoculture system (HTG: spatially heterogeneous; HMG: spatially homogeneous). Data were normalized to the AUC of the WT and log₁₀-transformed. Consequently, the AUC for the WT is 0 for each carbon source; values > 0 indicate an increase relative to WT, while values < 0 indicate a decrease. CreiNecro and EcoNecro denote the necromass of Chlamydomonas reinhardtii and Escherichia coli, respectively. The original AUC data are presented as beeswarm plots in Supplementary Figures 2-4.”

Comment:

Figure 4C – what is the y-axis here?

Response:

We apologize for the omission. We have updated the caption for Figure 4C to clarify that the y-axis represents the percentage of biospheres in each state (e.g., both species surviving, only one surviving, or neither) relative to the total number of biospheres harvested at that time point (see page 30 line 543-page 31 line 548).

“(C) The percentage of biospheres in each state relative to the total number harvested

per time point for the three treatments across the four harvests. “both” indicates biospheres where both E. coli and Chlamydomonas reinhardtii (C. reinhardtii) survived; “neither” indicates the opposite. “C. reinhardtii only” and “E. coli only” indicate biospheres where only the alga or the bacterium survived, respectively.”

Reviewer #4 (Remarks to the Author):

Comment:

In this study the authors carry out an evolution experiment with *E. coli* and *Chlamydomonas reinhardtii* in monoculture and co-culture. The monoculture phase propagated *E. coli* in heterogenous or homogenous conditions, and the coculture conditions involved taking the wt, HTG and HMG populations for coculture with *C. reinhardtii*. What makes this study unique is the experiments are carried out in closed microbial biospheres. Since this study is well connected with the microbial experimental evolution literature and theory, the results are discussed in the light of the expectations of the most relevant results of open system biology. Closed system biology is massively understudied, and this important contribution makes a significant advance. The findings highlight that a lot of work needs to be done to understand how ecological communities will evolve in closed systems. The authors are, hopefully, pioneers of a new field of robust microbial biosphere experiment and theory.

Response:

We are sincerely grateful for your positive assessment of our work and your encouraging words regarding the potential of closed system biology.

Comment:

The paper is very well written with almost no errors, the figures are beautifully presented. My comments below:

Figure 2. Panel D – should these biofilm measurements be normalised by the number of cells in the culture? I am concerned the increase in raw OD570 could be due to higher cell counts rather than greater per cell biofilm forming capacity.

Response:

Thank you for this insightful comment regarding the potential influence of planktonic cell density (OD600) on our biofilm measurement (OD570). We appreciate the opportunity to clarify our analytical approach.

To address your concern, we re-analyzed our raw data to evaluate the feasibility and validity of normalizing biofilm values by cell density (i.e., calculating an OD570/OD600 ratio). After careful consideration and statistical verification, we have decided to retain the original OD570 values in the manuscript for three key reasons.

First, we performed an Ordinary Least Squares (OLS) regression between blank-corrected OD600 (total cell density) and OD570 (biofilm biomass) across all samples. While the relationship was statistically significant, the coefficient of determination (R^2) was notably low (See Figure below). This indicates that variations in cell density explain only a small fraction of the variance in biofilm formation. In other words, high biofilm values are not simply a function of having more cells.

Correlation of blank-corrected OD600 to blank-corrected OD570 at three measurement time points, fitted using ordinary least squares (OLS) regression.

Second, when we attempted the suggested normalization (OD570/OD600), the resulting index reclassified nearly all our samples as “non-biofilm forming.” This result directly contradicted our qualitative, independent observations on Congo Red Agar (CRA) plates (see page 24 Table 1 and page 26 Figure 2B), where these same populations showed clear phenotypic evidence of biofilm production. This discrepancy likely arises because strong biofilm formers often have a higher proportion of cells embedded in the matrix (adhered to the wall surface) rather than in the planktonic phase, potentially leading to lower or non-linear OD600 readings. Normalizing by the planktonic optical density can therefore artificially mask the presence of the biofilm matrix.

Third, the crystal violet assay measures total biofilm biomass, which encompasses both bacterial cells and the secreted extracellular polymeric substance (EPS) matrix. Standard protocols in the field typically report OD570 values relative to negative controls rather than ratios. We have adhered to these established conventions to ensure our results are comparable with the broader literature.

<https://academic.oup.com/lambio/article/38/5/428/6703107>

<https://linkinghub.elsevier.com/retrieve/pii/S0362028X23042515>

<https://doi.org/10.1021/acsomega.9b03540>

Comment:

Line 222 I agree with the authors that slower divergence of *E. coli* W from its ancestor suggests that it experienced weaker selection pressures, however the explanation could be related to diminishing returns epistasis. Many evolution experiments have shown that in situations such as described here, where you have two different *e. coli* strains evolving in the same conditions for the same period of time, you would expect the more poorly adapted strain to evolve a greater increase in fitness during that period. The idea is that better adapted strains have less room for improvement. The authors mentioned that *E. coli* W has a wider carbon utilisation range, supporting that it was better adapted to the conditions of the experiment than the K12 derived strain. If the authors agree, perhaps they could mention this.

Response:

Thank you for this excellent theoretical insight. We fully agree that “diminishing returns epistasis” offers a compelling explanation for the behavior of strain W. Since this strain possessed a broader initial carbon utilization profile, it likely had higher initial fitness in the necromass environment, leaving less room for adaptive improvement compared to the K-12 derived MC4100 strain.

However, based on feedback from other reviewers and the editor regarding the complexity and length of the manuscript, we have made the strategic decision to remove all data and discussion related to the second strain (*E. coli* W) from the revised manuscript. As this strain showed minimal evolutionary response and was not used in the downstream co-culture biospheres, its inclusion was found to dilute the primary narrative. The revised manuscript now focuses exclusively on *E. coli* MC4100 to

provide a clearer, more robust analysis of how spatial structure shapes phenotypic evolution and subsequent ecological interactions.

Comment:

Line 230 The authors refer to “genomic streamlining” incorrectly - Genome streamlining is the evolutionary process where a genome reduces its size by shedding unnecessary genes to increase fitness. Here is a recent example of its use: <https://doi.org/10.1093/gbe/evae250>

Response:

Thank you for correcting our terminology. We agree that our use of “genome streamlining” was imprecise in that context. As noted in the response above, the section discussing the second strain (where this term appeared) has been removed from the revised manuscript. We have ensured that the remaining text uses precise evolutionary terminology.

Comment:

Line 292 I think that one of the most interesting findings is that the E. coli that had become better adapted to efficient nutrient cycling were not better able to enhance system persistence. Your possible explanation is not quite clear though – are you saying that since evolution in monoculture drove the loss of motility, this was the reason that the wt E. coli was a better “partner” for C. reinhardtii? This just needs some clarification in the writing.

Response:

Thank you for highlighting this critical point. We agree that the distinction between individual adaptation and ecological function needed to be sharper. We have revised the

Discussion to explicitly clarify this mechanism.

Our argument is that the monoculture environment imposed strong selection for energy conservation and surface attachment, driving the loss of motility and increased biofilm formation (as seen in the HMG populations). While these traits maximized *E. coli* survival in isolation, they created an “evolutionary mismatch” when reintroduced to the co-culture system. The wild-type progenitor, which retained motility, was likely better able to physically track the algae and access local exudate patches, facilitating the necessary two-way exchange of nutrients. By contrast, the “better adapted” (immotile) strains could not effectively couple with the producer, leading to faster system collapse despite their enhanced metabolic potential.

We have revised the text to clearly articulate this trade-off (see page 10 lines 208-220).

“Second, the loss of motility in pre-adapted E. coli likely disrupts spatial coupling with the producers. Bacterial motility facilitates proximity to C. reinhardtii, enabling efficient aerobic respiration and nutrient exchange^{38,39}. While motile populations (WT progenitor and HTG-derived populations HTG1/HTG2) could forage on C. reinhardtii exudates, HMG-derived populations lost motility and increased biofilm formation. This sessile lifestyle likely sequesters nutrients within the biofilm matrix. Since diffusion is a slow process, this sequestration privatizes and hoards shared nutrient resources (the “commons”) for immediate local benefit, while weakening interaction strength⁴⁰. This preferential resource retention limits reciprocal nutrient exchange with C. reinhardtii. Since algal proliferation depends on the accumulation of CO₂ and recycled nutrients⁴¹, this limitation reduces algal photosynthetic activity and potentially depletes oxygen availability. Consequently, due to this lack of oxygen, E. coli may be forced into less efficient anaerobic metabolism. These combined effects create a “tragedy of the commons” (wherein individual optimization undermines collective stability) and likely impede coexistence.”

Manuscript Title: “Rapid evolution in necromass use under resource limitation reduces persistence in producer-decomposer microbial biospheres”

We sincerely thank the reviewers for their positive assessment of our revised manuscript and the additional suggestions, which have enabled us to further improve the work. Below, we respond to the reviewers’ comments point by point.

Reviewer #1:

Comment:

It is appreciated that the authors have made substantial revisions to the manuscript to address concerns regarding the study’s model, data visualization, and conclusions. These revisions have significantly improved the quality of the study. However, given the extent of the changes, we still have several remaining concerns, as well as a few additional issues arising from the revised text.

To address the concerns regarding the model, the authors now explicitly stated in the manuscript that *C. reinhardtii* can compete with *E. coli* for necromass and that the loss of fluorescence reflects a collapse of primary production rather than necessarily indicating extinction. This distinction is important as it highlights key caveats that are critical to interpreting the study’s conclusions.

The authors have also revised the manuscript to clearly state that the WT *E. coli* strain demonstrates greater longevity than the evolved strains and that the initial hypothesis is not supported by the data. We agree with the authors that the removal of the *E. coli* W strain improves the focus of the key findings of the study.

It is also appreciated that the authors made efforts to improve readability by substantially revising the introduction and condensing the discussion. The addition of

literature discussing LTSP and GASP helps better situate the study within the current state of the field. However, the revisions also introduce several issues related to metabolism that should be addressed. These concerns are outlined below:

Line 52-53: “*E. coli* respire CO₂ from necromass decomposition for photosynthetic assimilation by *C. reinhardtii*, which in turn maintains aerobic conditions via O₂ release.” This needs to be reworded because, as written, it initially sounds like the authors are claiming *E. coli* is carbon-fixing (respiring CO₂), which they cannot do. It is also unclear if there is evidence that *E. coli* aren’t fermenting necromass under these conditions. It would be more accurate to write: *E. coli* metabolizes necromass, releasing CO₂ for photosynthetic.....

Response:

We appreciate your positive feedback on the revised manuscript and for noting this ambiguous phrasing. We agree that the original wording was misleading and might imply that *E. coli* performs carbon fixation. We have revised the sentence (see page 3 lines 51-53). The revision now more accurately captures the decomposer role of *E. coli* in the system.

“E. coli metabolizes necromass, releasing CO₂ for photosynthetic assimilation by C. reinhardtii, which in turn maintains aerobic conditions via O₂ release.”

Line 211-212 and 220-221: Fermentation should be mentioned as a possibility here as well.

Response:

Thank you for this helpful suggestion. We agree that fermentation should be acknowledged as a possible metabolic alternative under oxygen-limited conditions. We have revised the relevant passages in the Discussion to explicitly mention fermentation as a possibility: (1) in the context of incomplete necromass degradation under low-oxygen conditions (see page 10 lines 205-206), and (2) when describing the potential consequences of reduced photosynthetic activity on *E. coli* metabolism (see page 10 lines 219-220).

*“Although starved *E. coli* incompletely metabolizes endogenous compounds and necromass³⁶, and may resort to fermentation under oxygen-limited conditions, potentially releasing metabolites such as aromatic amino acids (e.g., phenylalanine) that could be utilized by *C. reinhardtii*³⁷, its aggressive resource acquisition likely overwhelms this potential benefit and may suppress *C. reinhardtii*.”*

*“Consequently, due to this lack of oxygen, *E. coli* may be forced into less efficient anaerobic metabolism, relying primarily on the fermentation of necromass.”*

The revisions to the manuscript figures help in improving the clarity and effectiveness of the data visualization. With that being said, there are still remaining concerns.

The use of the Sankey diagram in Figure 2B remains uninformative as it does not allow tracking of populations through the diagram. If the goal is to show phenotypic shifts, the data cannot be combined for each trait. Each track must should remain separate (ie. separate positive and negatives for each treatment, WT, HTG, and HMG). The table is far more practical and it is recommended that the Sankey diagram be eliminated from the final version.

Response:

We agree that the Sankey diagram in Figure 2B was not effectively conveying the

phenotypic shifts across populations and traits. Therefore, we have removed this panel. The figure caption and all in-text references have been updated accordingly (see page 26 Figure 2).

Finally, the use of the qualitative method for measuring motility is not ideal, particularly when analyzing evolved phenotypes in a small number of populations. Nevertheless, given that the data are collected, it is recommended that the analysis proceeds but with the limitation acknowledged.

Response:

Thank you for this constructive comment regarding the qualitative motility assay. We have added an acknowledgement of this limitation to Discussion 3.3 Limitations and future perspectives (see page 13 lines 271-272).

“Third, motility was assessed using a qualitative binary method, which may not capture the full continuum of motility phenotypes.”

Reviewer #2:

Comment:

I co-reviewed this manuscript with one of the reviewers who provided the listed reports. This is part of the Communications Biology initiative to facilitate training in peer review and to provide appropriate recognition for Early Career Researchers who co-review manuscripts.

Response:

Thank you for your valuable input.

Reviewer #3:

Comment:

The authors have addressed all comments. Thanks to the authors for their thoughtful responses!

Response:

Thank you for your constructive comments that have enabled us to improve this manuscript.

Reviewer #4:

Comment:

The authors have addressed all of my concerns, and I have no further comments. Congratulations on a really nice study.

Response:

We truly appreciate your recognition of our work.